# Semiconductor SERS enhancement enabled by oxygen incorporation

Zuhui Zheng[1,2], Shan Cong[1], Wenbin Gong[1], Jinnan Xuan[3], Guohui Li[3], Weibang Lu[1], Fengxia Geng[3] & Zhigang Zhao[1]

Semiconductor-based surface-enhanced Raman spectroscopy (SERS) substrates represent a new frontier in the field of SERS. However, the application of semiconductor materials as SERS substrates is still seriously impeded by their low SERS enhancement and inferior detection sensitivity, especially for non-metal-oxide semiconductor materials. Herein, we demonstrate a general oxygen incorporation-assisted strategy to magnify the semiconductor substrate–analyte molecule interaction, leading to significant increase in SERS enhancement for non-metal-oxide semiconductor materials. Oxygen incorporation in $MoS_2$ even with trace concentrations can not only increase enhancement factors by up to 100,000-fold compared with oxygen-unincorporated samples but also endow $MoS_2$ with low limit of detection below $10^{-7}$ M. Intriguingly, combined with the findings in previous studies, our present results indicate that both oxygen incorporation and extraction processes can result in SERS enhancement, probably due to the enhanced charge-transfer resonance as well as exciton resonance arising from the judicious control of oxygen admission in semiconductor substrate.

[1] Key Lab of Nanodevices and Applications, Suzhou Institute of Nano-Tech and Nano-Bionics Chinese Academy of Sciences (CAS), Suzhou 215123, China. [2] Nano Science and Technology Institute, University of Science and Technology of China, Suzhou 215123, China. [3] College of Chemistry, Chemical Engineering and Materials Science, Soochow University, Suzhou 215123, China. Correspondence and requests for materials should be addressed to Z.Z. (email: zgzhao2011@sinano.ac.cn)

Surface-enhanced Raman spectroscopy (SERS) is a sensitive analytical technique that is capable of detecting and identifying chemical and biological compounds with single-molecular sensitivity through their unique vibrational fingerprints, which opens the way for a wide variety of applications such as forensics, homeland security, food safety, and medical diagnostics[1–3]. In SERS measurements, substrate material is indispensable because the huge enhancement effect can be obtained from substrate material through two well-known mechanisms, i.e., electromagnetic mechanism (EM) and chemical mechanism (CM)[4]. EM is a mechanism based on the enhancement of the local electromagnetic field as a result of the surface plasmons excited by the incident light, while CM is mostly believed to be due to a charge transfer between the molecule and the substrate[5]. Traditionally, noble metals such as Au, Ag, and Cu have been dominant as the substrate materials for SERS mainly due to the existence of EM. However, there are some disadvantages to use noble metals for working as SERS substrates, such as high cost, poor uniformity, lack of stability, and biocompatibility, side-reactions of the adsorbate due to the catalytic effect of metals, which seriously restrict the utilization of noble-metal SERS substrates in practical applications[6]. In contrast, semiconductor SERS substrates offer not only higher SERS uniformity but also better chemical stability and biocompatibility[7,8]. Additionally, the effective utilization of semiconductor SERS substrates might lead to a more enlightened and thereby broadened use of SERS in many fields, such as direct monitoring of interfacial chemical reactions on individual nanoparticles[9]. Thus, in the past few years, semiconductor SERS substrates have gained immense popularity, such as InAs/GaAs quantum dots[10], CuTe nanocrystals[11], and $TiO_2$ nanostructures[12,13], in which CM effect plays a major role in Raman scattering enhancement. Unfortunately, compared with noble metals (on the order of $10^6$–$10^{10}$ and $10^{-10}$ M), the enhancement factor (EF) and limit of detection (LOD) of semiconductor SERS substrates are generally quite low, usually in the range of 10–$10^2$ and $10^{-3}$ M, respectively, far from sufficient for application in various chemical and biological sensing. Therefore, exploring novel strategies to greatly improve the SERS performance of semiconductor substrates has become an urgent task[7,14].

In 2015, we have proposed that making oxygen vacancies is a powerful means for improving the SERS enhancement of semiconductor metal oxide based on the charge transfer between the substrate and the adsorbed molecules. It is found that oxygen vacancies can increase the EF of tungsten oxide[7], which is previously considered to be non-SERS-active, to as high as $3.4 \times 10^5$. Further, a recent breakthrough made by Guo et al. has transformed non-SERS or weak $Cu_2O$, ZnO, and $MoO_3$ substrates into highly SERS-active substrates with EFs as high as $1.8 \times 10^7$ and LODs as low as $10^{-9}$ M by oxygen vacancy engineering, respectively[8,15,16]. Specially, Wang et al.[16] indicate that the SERS activity of amorphous ZnO nanocages can be attributed to the high-efficiency interfacial charge transfer, which is assisted by the numerous metastable electronic states of amorphous ZnO nanocages. Xi et al.[17] have also reported that oxygen vacancy-rich $MoO_2$ can be used as a sensitive SERS substrate to detect trace amounts of highly risk chemicals including bisphenol A, dichlorophenol, pentachlorophenol, and so on. Zhao et al. have also reported the ultra-sensitive detection of benzoic acid analogs on $TiO_2$-based SERS substrates with LODs as low as $10^{-8}$ M, which takes advantage of the promoted charge transfer between the substrate and adsorbed molecules via oxygen vacancies[18–20]. These results indicate that the introduction of oxygen vacancies can be regarded as an exciting strategy to design high-performance semiconductor SERS substrates. However, this strategy can only be utilized in application for metal oxide materials, and new strategies for the development of high-performance semiconductor SERS substrates are still needed for other non-metal-oxide semiconductors.

Oxygen incorporation is a structural/electronic modulation strategy of semiconductor materials, which is the inverse process of making oxygen vacancies. As a matter of fact, oxygen incorporation has been expected to be capable of dramatically altering material properties even when such compositional changes are minute[21]. Thus, oxygen incorporation process is of key importance for a variety of applications, for example, the incorporation of lattice oxygen can enhance catalyst's activity and selectivity (Mars–van Krevelen mechanism)[22,23], oxygen incorporation is the signal-determining process in bulk conductivity sensors[24], and oxygen incorporation also plays a crucial role for mixed conducting cathodes of solid oxide fuel cell[25,26]. Synergistic structural and electronic modulations by oxygen incorporation should also be of particular interest for semiconductor SERS, based on the possibility of significantly enhanced SERS activity. However, to the best of our knowledge, this oxygen incorporation strategy has not yet been reported for semiconductor SERS.

Herein, using $MoS_2$ as a model material, we put forward that oxygen incorporation in $MoS_2$ can not only increase EFs by up to 100,000 times compared with oxygen-unincorporated samples but also endow $MoS_2$ with low LODs below $10^{-7}$ M. What is more interesting is that the EFs of $MoS_2$ continuously increase to a maximum value with increasing oxygen incorporation concentrations as long as its phase structure remains undisrupted; however, it quickly drops to a very small value when the phase change is occurring. Besides, this approach is also applicable to other non-metal-oxide semiconductors, such as $WS_2$ and $MoSe_2$, thus demonstrating the universality of this approach. Finally, based on the simulation and experimental results, a SERS model is proposed to explain why both oxygen incorporation and extraction can result in SERS enhancement.

## Results

**Synthesis and characterizations.** Structurally, oxygen incorporation in $MoS_2$ has two fundamentally different types: substitution of oxygen for sulfur within $MoS_2$ lattices and partial oxidation of molybdenum atoms along edge planes or at defects of $MoS_2$ crystals[27]. The former one involves oxygen substituted into the trigonal prisms of the $MoS_2$ hexagonal lattice forming a solid solution, $MoS_{2-x}O_x$ phase (Fig. 1b), while in the latter case, oxidation occurs on the edge planes or at defects of $MoS_2$ crystals such as steps, kinks, or vacancies in the basal surfaces, and produces a very small amount of new compound, $MoO_3$, around the edges or defects (Fig. 1c). The significant difference between substitution and oxidation is that Mo(VI) can be observed in the partial oxidation process instead of in the oxygen substitution process[27]. On the basis of the two structures above, two synthetic routes are accordingly used to prepare oxygen-incorporated $MoS_2$. For oxygen-substituted $MoS_2$, low-temperature hydrothermal treatment of $(NH_4)_6Mo_7O_{24}\cdot4H_2O$ with the assistance of thiourea as an additive is performed using an autoclave at a temperature of 200 °C for 20 h. The lower reaction temperature causes the reaction process to be insufficient, leading to remaining Mo–O bonds inherited from the molybdate precursor, thus realizing the oxygen substitution (Fig. 1a). For partially oxidized $MoS_2$, the preparation is achieved through annealing of $MoS_2$ sample at temperatures varying from 250 to 300 °C for a short span of time in air (Fig. 1c) (the details on synthesis in the Supplementary Methods).

The hydrothermally treated oxygen-substituted $MoS_2$ samples are first characterized with respect to physical and structural properties using X-ray diffraction (XRD), X-ray photoelectron

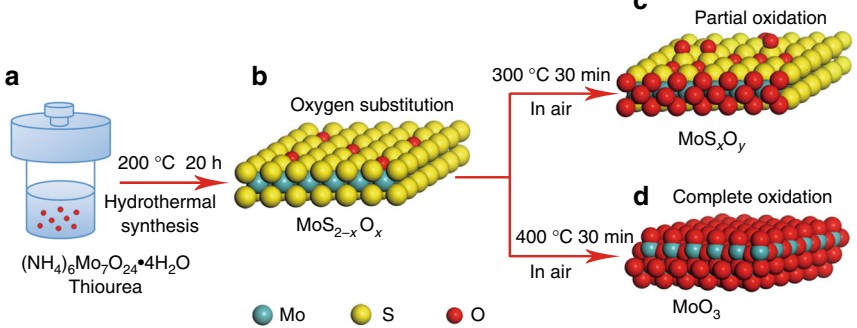

**Fig. 1** Structure and synthesis of oxygen-substituted and oxidized MoS$_2$. Oxygen incorporation in MoS$_2$ has two fundamentally different types: substitution of oxygen for sulfur within MoS$_2$ lattices and partial oxidation of molybdenum atoms along edge planes or at defects of MoS$_2$ crystals. **a, b** Low-temperature hydrothermal treatment of (NH$_4$)$_6$Mo$_7$O$_{24}$·4H$_2$O with the assistance of thiourea as an additive realizes oxygen substitution. **c** Annealing of MoS$_2$ sample at 300 °C for 30 min produces partially oxidized MoS$_2$ without changing its crystal structure. **d** Annealing of MoS$_2$ sample at 400 °C for 30 min makes MoS$_2$ completely oxidized to MoO$_3$, which reshapes its crystal structure

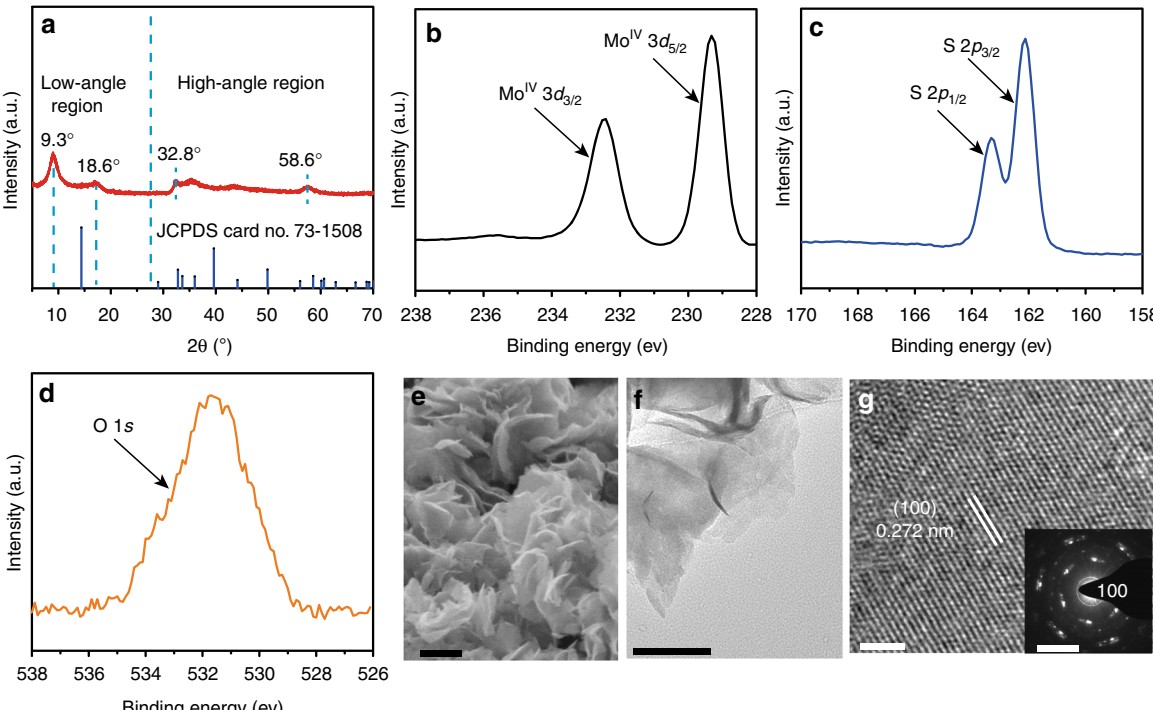

**Fig. 2** Morphology and structural characterizations of the oxygen-substituted MoS$_2$. **a** XRD pattern of the hydrothermally treated oxygen-substituted MoS$_2$, and the standard pattern of the pristine 2H–MoS$_2$ (JCPDS No. 73-1508). XPS spectra showing the binding energies of **b** molybdenum, **c** sulfur, and **d** oxygen. **e** FE-SEM image, **f** TEM image, and **g** high-resolution TEM image of the oxygen-substituted MoS$_2$. Inset of **g** is the selected area electron diffraction pattern of the oxygen-substituted MoS$_2$. Scale bars: **e** 200 nm; **f** 100 nm; **g** 2 nm. Inset of **g**, 5 1/nm

spectra (XPS), scanning electron microscopy (FE-SEM), and transmission electron microscopy (TEM). Figure 2a shows the XRD pattern of oxygen-substituted MoS$_2$ sample after hydrothermal treatment at 200 °C. Similar to that of the pristine 2H–MoS$_2$ (JCPDS 73-1508), two broadened peaks at high-angle region (32.8° and 58.6°) are clearly observed, which can be well indexed to (100) and (110) planes of the pristine 2H–MoS$_2$, indicating the same atomic arrangement along the basal planes[28]. However, the situation becomes different at the low-angle region. The XRD pattern of the pristine 2H–MoS$_2$ usually shows a single strong (002) reflection at 14.4° (corresponding to a d-spacing of 6.15 Å), which is the dominant peak for a well-stacked layered MoS$_2$ crystal[29]. On the contrary, for our oxygen-substituted MoS$_2$ sample, two new peaks at 9.3° and 18.6° with diploid relationship

appears at low-angle region corresponding to (001) and (002) reflections with d-spacings of 9.7 and 4.8 Å, respectively. The significantly expanded interlayer spacing of our samples is presumably a result of oxygen incorporation. As we know, the oxygen is more electronegative than sulfur, which means oxygen incorporation could result in larger repulsive interactions between adjacent layers and subsequently cause the large expansion of interlayer spacing of MoS$_2$[28]. XPS analysis is used to probe the chemical state and composition of the hydrothermally treated sample. The XPS survey spectra clearly suggest that the sample mainly consists of three elements: Mo, S, and O. The high-resolution Mo 3d peak can be fitted into only two doublets arising from Mo 3d$_{5/2}$ and Mo 3d$_{3/2}$ orbitals located at 229.3 and 232.4 eV (Fig. 2b), suggesting the +4 oxidation state of Mo (Mo(IV)) is

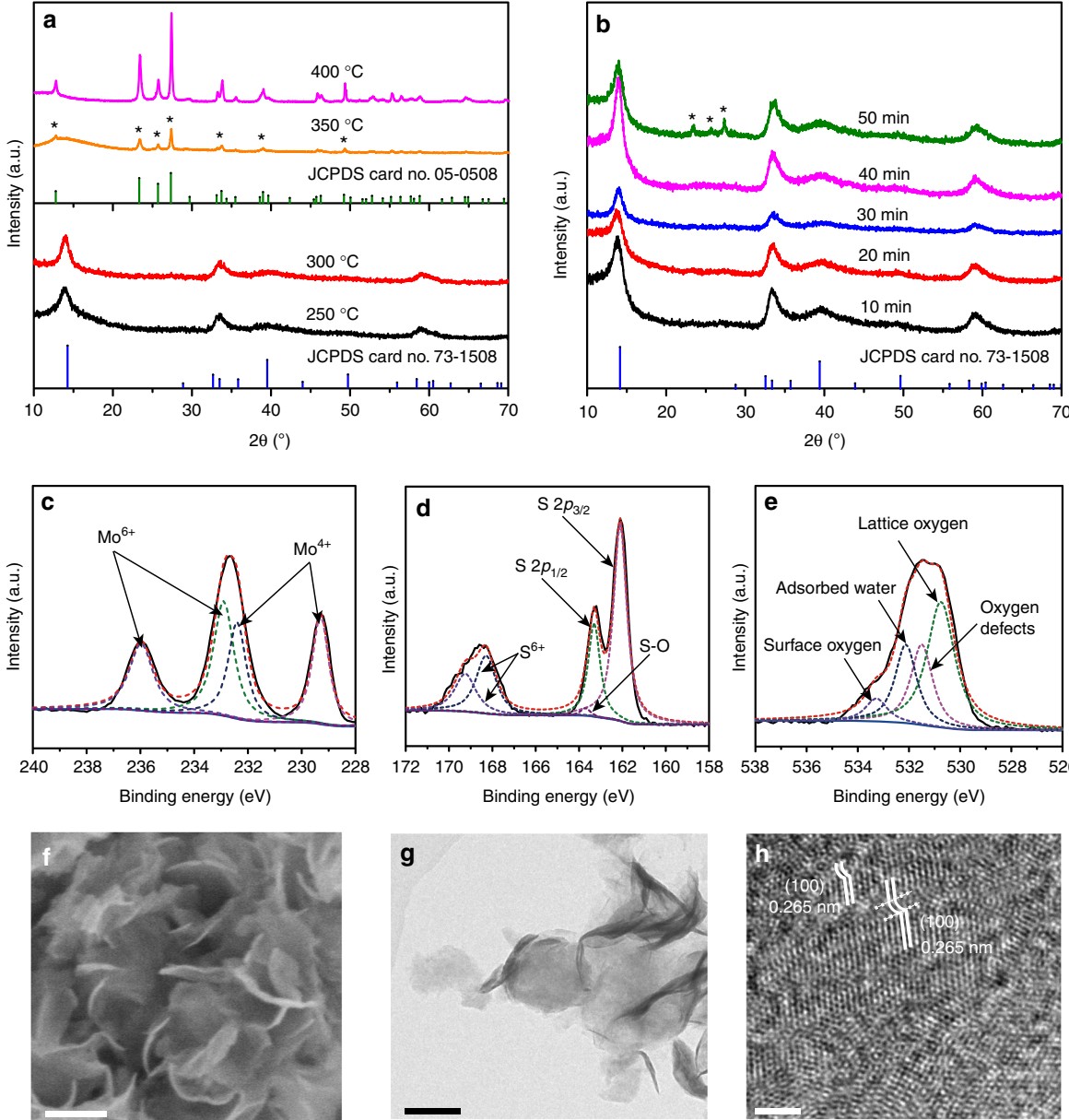

**Fig. 3** Morphology and structural characterizations of oxidized $MoS_2$. XRD patterns of the oxidized samples obtained at **a** various annealing temperatures and **b** various annealing times, and the standard patterns of the pristine $2H-MoS_2$ (JCPDS No. 73-1508) and $MoO_3$ (JCPDS No. 05-0508). XPS spectra showing the binding energies of **c** molybdenum, **d** sulfur, and **e** oxygen. **f** FE-SEM image, **g** TEM image, and **h** high-resolution TEM image of the $MoS_2$ sample annealed at 300 °C for 40 min. Scale bars: **f**, **g** 100 nm; **h** 2 nm

dominant for the hydrothermally treated sample[30]. The XPS spectra in the S 2p and O 1s regions are also displayed in Fig. 2c, d, respectively. The deconvolution of the S 2p peak illustrates the presence of S $2p_{3/2}$ and S $2p_{1/2}$ spin-orbit doublet, which can be indexed to Mo–S bonding in $MoS_2$. Furthermore, the O 1s region shows a characteristic peak located at 530.0 eV corresponding to the binding energy of oxygen in $Mo(IV)O_2$, suggesting the existence of Mo(IV)–O bonds, thus verifying the successful oxygen incorporation in $MoS_2$[31]. FE-SEM and TEM are also performed on the hydrothermally treated oxygen-substituted $MoS_2$ sample. SEM and TEM images outline that the sample composes of crumpled and entangled nanosheets with uniform lateral size in the range of 100–200 nm (Fig. 2e, f). Selected area electron diffraction analysis indicates that the nanosheets are crystalline in nature. High-resolution TEM (HRTEM) image of the nanosheets indicate the interplanar spacing is about 0.272 nm,

which corresponded to the (100) crystallographic plane, in accordance with XRD results (Fig. 2g).

Next, the hydrothermally treated samples at 200 °C continue to be used as precursors to prepare partially oxidized $MoS_2$ samples. Figure 3a shows the XRD spectra of the partially oxidized $MoS_2$ samples prepared by oxidizing the precursor in air for 30 min at temperatures of 250, 300, 350, and 400 °C, respectively. The precursor before heat treatment barely exhibit the characteristic (002) peak of the layered $MoS_2$ crystal (Fig. 2a), while the XRD patterns of the samples annealed at 250 and 300 °C in Fig. 3a can readily be indexed as hexagonal $2H-MoS_2$ (JCPDS 73-1508), with lattice parameters $a = 3.962$ Å, $b = 13.858$ Å and $c = 3.697$ Å. The single strong (002) reflection at $2\theta = 14.4°$, corresponding to a d-spacing of 0.62 nm, indicates a well-stacked layered structure of the annealed $MoS_2$ samples. The other three peaks located at 32.8, 39.7, and 58.6° can also be assigned to (100), (103), and (110)

reflection of 2H–MoS$_2$, respectively, demonstrating the formation of 2H–MoS$_2$. Additionally, it is noticeable that no other phases except 2H–MoS$_2$ are detected in the annealed samples as the temperature of oxidation is increased from 250 to 300 °C. However, when the oxidation is carried out at 350 °C, significant changes are observed in the XRD patterns of the annealed samples. Additional peaks (marked by *) associated with the orthorhombic phase of MoO$_3$ (JCPDS 05-0508) can be clearly distinguished at 2θ values of 12.6°, 23.3°, 25.7°, 27.3°, 34.4°, 39.7°, and 49.2°, which correspond to (020), (110), (040), (021), (140), (150), and (002) reflections of MoO$_3$, respectively. Upon annealing at 400 °C MoS$_2$ is completely converted to crystalline MoO$_3$ without other phases. The type of phase transition is also quite dependent on oxidation time, as shown in Fig. 3b. Based on the XRD results, the samples annealed at 300 °C only contain 2H–MoS$_2$ phase when the oxidation time is <40 min, while a set of XRD signals arising from MoO$_3$ appears at 23.3°, 25.7°, and 27.3° as the oxidation time is increased to 50 min. Although MoO$_3$ phase occurs in XRD patterns only at higher oxidation temperature and longer oxidation time, XPS demonstrate that high amounts of oxygen have been incorporated into all the annealed samples (Fig. 3c–e, Supplementary Fig. 2). Figure 3c–e shows the XPS spectra of the Mo 3d, S 2p, and O 1s core level in the sample annealed at 300 °C for 40 min. The deconvolution of Mo 3d spectrum reveals two Mo 3d doublets: the fitted Mo 3d peaks positioned at 232.18 and 235.40 eV are corresponding to Mo(6+) oxidation state[8], and the doublet peak at 229.3 and 232.4 eV belongs to Mo(4+) oxidation state (Fig. 3c). The change in the oxidation number of molybdenum atoms from +4 to +6 provides compelling evidence that oxygen incorporation by annealing is operated in oxidation mode. In addition, the change of oxidation number is also observed for sulfur. A new doublet with S 2p$_{3/2}$ and S 2p$_{1/2}$ components, adjacent to the original S 2p position, appears at 168.6 and 170.1 eV (Fig. 3d), giving information that parts of surface sulphur are oxidized to the +6 state by the annealing process[32]. Additionally, a weak S 2p doublet at 163.7 eV

corresponding to S–O bonding is detectable due to the chemical adsorption of oxygen atoms on the topmost of S atoms in MoS$_2$ after 300 °C annealing[33]. On the other side, as a result of oxidation, an extra asymmetric and broad O 1s peak is observed on all samples after annealing at 532 eV (Fig. 3e, Supplementary Fig. 2). After deconvolution, it is found that oxygen has different origins: the peak centered at 530.3 eV is associated with the O$^{2-}$ ions in the metal oxide (lattice oxygen), the peak at 531.4 eV is attributed to the O$^{2-}$ ions in the oxygen-deficient regions, and the binding energy peaks at 530.9 and 532.0 eV are likely due to the surface loosely bound O$_2$ or adsorbed water, respectively[28,34]. XPS analysis also gives the changing trends of oxygen incorporation concentrations as a result of temperature and time (Supplementary Fig. 3). The oxygen incorporation concentration increases from 44.81 to 89.64% when the annealing temperature is raised from 250 to 350 °C. Similarly, as the annealing time is increased from 10 to 50 min, the amount of incorporated oxygen also increases from 39.91 to 67.95%. Notably, non-stoichiometric products during oxygen incorporation are obtained as illustrated by the XPS analysis, with the O/Mo ratio exceeding 2 (totally substituted MoS$_2$) or even 3 (totally oxidized MoO$_3$). The origin of this observation is probably attributed to the existence of chemically bound oxygen on the topmost S layer and surface-adsorbed sulfoxides in MoS$_2$ crystals, similar to those observed in previous reports[32,35]. Although large quantities of oxygen have been incorporated into MoS$_2$ lattice, it is found that the nanosheet morphology is quite well preserved even after calcination at 300 °C for 40 min (Fig. 3f, g). However, oxygen incorporation can cause different degrees of lattice distortion in the MoS$_2$ hosts, as depicted by the HRTEM photographs of the partially oxidized samples (Fig. 3h). In such distorted regions, local electronic properties would be significantly altered from those obtained in the undistorted regions (as validated by our calculation results of Bader charges, Supplementary Fig. 4)[28], probably impacting on the charge-transfer efficiency and magnifying the molecular polarization, ultimately resulting in enhanced SERS signals for

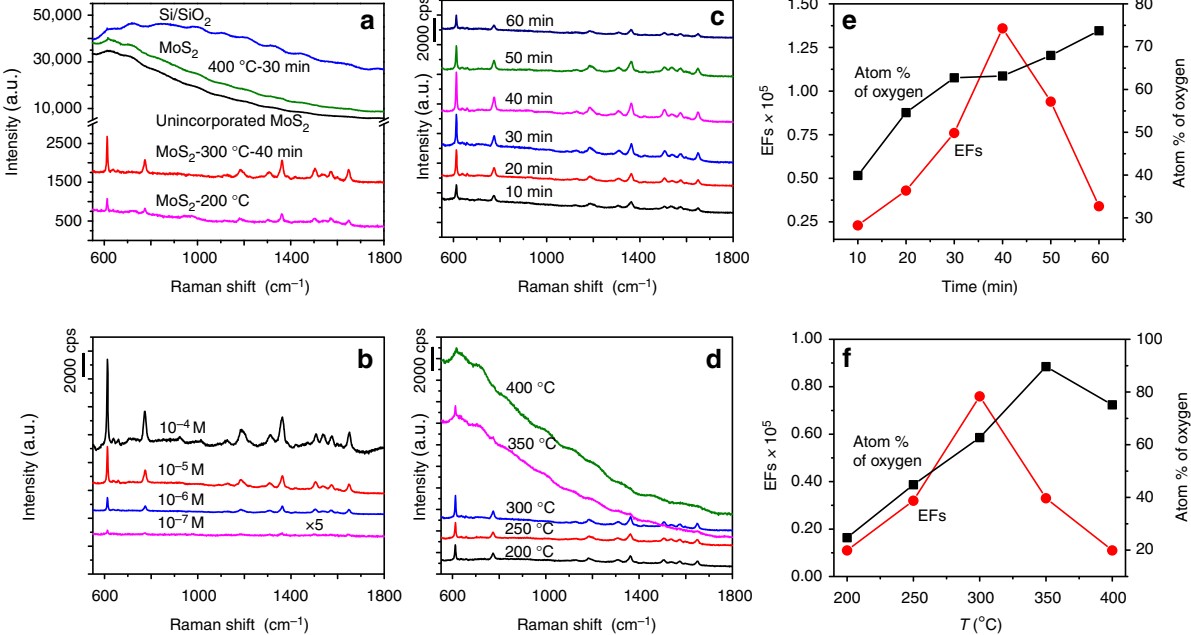

**Fig. 4** SERS properties. **a** Raman profile of R6G (10$^{-6}$ M) on substrates deposited with unincorporated MoS$_2$ sample, hydrothermally treated oxygen-substituted MoS$_2$ sample at 200 °C, partially oxidized sample at 300 °C for 40 min, completely oxidized MoO$_3$ sample, and bare SiO$_2$/Si. **b** Raman spectra collected for oxygen-incorporated MoS$_2$ sample annealed at 300 °C for 40 min at four different concentrations, 10$^{-4}$, 10$^{-5}$, 10$^{-6}$, and 10$^{-7}$ M, suggesting the detection limit was as low as 10$^{-7}$ M. SERS spectra on the oxygen-incorporated MoS$_2$ samples with different annealing times (**c**) and temperatures (**d**). **e**, **f** The dependence of the SERS enhancement of R6G on the oxygen incorporation concentration for both types of oxygen-incorporated MoS$_2$ samples

molecules attached to them, just as demonstrated in amorphous ZnO reported by Guo's group[16].

**SERS properties of oxygen-incorporated MoS₂ samples.** Very interestingly, the unincorporated MoS₂ samples, oxygen-substituted MoS₂ samples, partially oxidized MoS₂ samples, and completely oxidized samples are found to exhibit strong differences in the SERS behaviors, as shown in Fig. 4. The SERS behaviors of these materials are examined using rhodamine 6G (R6G) as a target molecule, the molecular structure of which is provided in Supplementary Fig. 5. Figure 4a shows the Raman spectra of R6G ($10^{-6}$ M) with the excitation wavelength being 532.8 nm on substrates deposited with unincorporated MoS₂ sample, hydrothermally treated oxygen-substituted MoS₂ sample at 200 °C, partially oxidized sample at 300 °C for 40 min, completely oxidized MoO₃ sample, along with bare SiO₂/Si. A key observation is that before oxygen incorporation, the SERS spectra of pristine, unincorporated MoS₂ sample, and completely oxidized MoO₃ sample only contain a broad fluorescence background but rather weak R6G signals, while both oxygen-substituted and partially oxidized MoS₂ samples can give quite obvious SERS signals of R6G molecules. Several prominent bands originated from R6G molecules are found at 612, 773, 1360, and 1650 cm$^{-1}$, which can be assigned to in-plane and out-of-plane bending motions of carbon, hydrogen atoms of the xanthene skeleton, and aromatic C–C stretching vibration modes, respectively[7]. In order to quantitatively compare SERS activities for the above three samples, EF calculations are performed based on the magnification of Raman intensity compared with that on bare substrate (details in Supplementary Methods). Encouragingly, the EFs can reach as high as $1.1 \times 10^4$ and $1.4 \times 10^5$ for oxygen-substituted and partially oxidized MoS₂ samples calculated with the intensity of 612 cm$^{-1}$

band, respectively, which is about 10,000 and 100,000 times higher than that for pristine, unincorporated MoS₂ sample. SERS measurements are also performed for the partially oxidized sample at 300 °C for 40 min under different concentrations of R6G molecules (the range was selected according to adsorption isotherms in Supplementary Fig. 6), decreasing from $10^{-4}$, $10^{-5}$, and $10^{-6}$ to $10^{-7}$ M (Fig. 4b). Even the concentration of R6G solution is decreased to $10^{-7}$ M, the SERS signals are still conspicuous, thus the detection limit for partially oxidized MoS₂ sample can be determined to be $10^{-7}$ M, which is obviously higher than most previous studies of semiconductor SERS substrates[13,36–39]. Afterwards, the SERS spectra of R6G on partially oxidized MoS₂ samples with different annealing times and temperatures (it means that the stoichiometry or oxygen incorporation concentrations of MoS₂ should be changed) are shown in Fig. 4c, d. Annealing at 300 °C for <40 min can continuously and significantly enhances SERS signals, but continuing the annealing for longer time largely suppresses SERS signals. For example, for the sample annealed at 300 °C for 3 h, the SERS signals of R6G almost disappear (Supplementary Fig. 7). Similarly, SERS signals can be enhanced by lower annealing temperatures in the range of 200–300 °C, but is largely degraded by higher annealing temperatures. Additionally, it is noticeable that R6G adsorbed on the sample annealed at 400 °C, which has been mainly identified as MoO₃ (Fig. 3a), has a strong fluorescence background, whereas no fluorescence background from the samples annealed at lower temperatures is observed. This strongly supports the view that partial oxidation at low annealing temperatures is useful for not only enhancing SERS activity but also suppressing fluorescence background, which may be associated with the generation of more free carriers by oxygen incorporation. Figure 4e, f shows the dependence of the SERS enhancement of R6G on the oxygen incorporation concentration for both types of oxygen-incorporated MoS₂ samples. As shown in

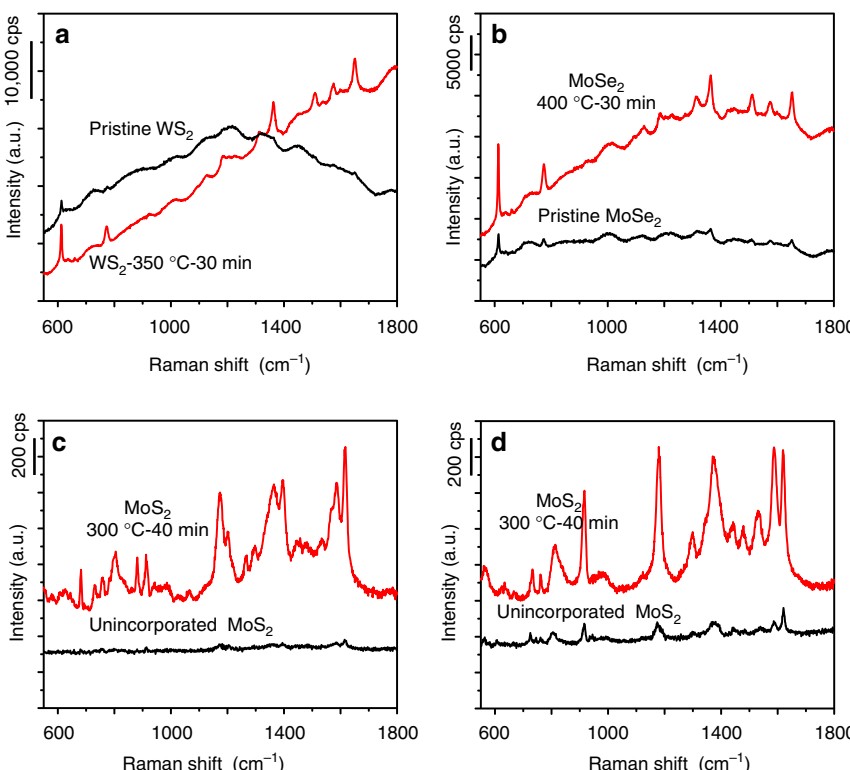

**Fig. 5** The universality of oxygen incorporation approach. The approach can be extended to other materials for a series of probe molecules. SERS spectra of R6G on a series of other semiconductor materials **a** WS₂ and **b** MoSe₂. SERS spectra of Victoria blue B **c** and crystal violet **d** on the partially oxidized MoS₂ sample at 300 °C for 40 min and unincorporated MoS₂, respectively

Fig. 4e, the EFs are found to be first increased with increasing oxygen incorporation concentration, reaches the maximum value at a specific oxygen incorporation concentration of 63.17%, and then drops sharply to almost zero with further increasing the oxygen incorporation concentration to 67.95%. Very surprisingly, even minor changes in oxygen incorporation concentration can induce dramatic changes in the SERS performance of $MoS_2$.

**Good universality and expandability.** Attractively, our approach is generically applicable to other types of non-metal-oxide semiconductors such as $WS_2$ and $MoSe_2$ (Fig. 5a, b). When using commercial 2H-phase $WS_2$ and $MoSe_2$ samples as the substrate materials, relatively weak SERS signals are detected. To amplify the weak SERS signals, annealing of $WS_2$ and $MoSe_2$ samples in air at temperatures of 350 and 400 °C for 30 min is performed, respectively. Similarly, the EFs for the annealed $WS_2$ and $MoSe_2$ samples can reach as high as $1.2 \times 10^5$ and $1.0 \times 10^5$, respectively. The SERS performances of these oxygen-incorporated samples are also in the rank of the high sensitivity among semiconducting materials, even comparable to noble metals without hot spots.

Further investigation demonstrates that the good universality can also be verified for other probe molecules. Specifically, in addition to R6G, the other trace-level probe molecules such as Victoria blue B and crystal violet can also be determined even at an extremely low concentration. Molecular structures of those molecules are illustrated in Supplementary Fig. 8. For Victoria blue B on the partially oxidized $MoS_2$ sample at 300 °C for 40 min, the EFs can reach as high as $1.4 \times 10^4$, and the detection limit can be determined to be $10^{-5}$ M (Fig. 5c). In sharp contrast, the EF is only 380 for Victoria blue B on the pristine $MoS_2$ sample. As for crystal violet, the partially oxidized $MoS_2$ sample yields more excellent SERS performance. The EFs on the partially oxidized $MoS_2$ sample is up to about $1.6 \times 10^5$, and the detection limit can be as low as $10^{-6}$ M (Fig. 5d). This may result from a relatively stronger interaction between the partially oxidized $MoS_2$ sample and crystal violet molecule.

**The mechanism of oxygen incorporation-assisted SERS enhancement.** In our recent studies, it has been established that oxygen vacancies play an important role in enhancing semiconductor SERS effect[7]. The present findings unexpectedly show that the inverse process of making oxygen vacancies, oxygen incorporation, could also effectively magnify the SERS signals of semiconductor materials. Taken together, it is interesting to see whether the SERS enhancement effect induced by oxygen vacancies and oxygen incorporation could reply on the same mechanism. Taking its cue from the pioneering theory of Lombardi et al.[40] on the SERS mechanism related to semiconductor materials, we consider if some resonances such as charge transfer, exciton, and molecular resonances are involved in the mechanism.

Charge-transfer resonance is a resonance Raman-like process associated with the photon-induced charge transfer from the semiconductor band edges to the affinity levels of the adsorbed molecule. This results in a change of the polarizability of the molecule, and consequently amplifies the intensity of its Raman signal[15]. For our partially oxidized $MoS_2$ samples, there are considerable experimental evidences for charge transfer through vibronic coupling. Note, for example, in Fig. 4, the lines at 612 and 773 $cm^{-1}$ (corresponding to in-plane and out-of-plane bending motion of the hydrogen atoms of the xanthene skeleton, respectively) can be seen to be the most enhanced lines in the spectra. These lines are well-known to be vibronically coupled[41], and therefore tend to be highly enhanced in SERS wherever charge transfer is important. Comparative analysis of the energy-level structures of pristine $MoS_2$, partially oxidized $MoS_2$, fully oxidized $MoO_3$, and R6G further indicates that partially oxidized $MoS_2$ provides significant advantages over other samples in charge transfer. As depicted in Fig. 6, when R6G is used as the target molecule, its HOMO (highest occupied molecular orbital) and LUMO (lowest unoccupied molecular orbital) levels are at −5.7 and −3.4 eV, respectively. Examining the energy levels of the above three semiconductors, we find that the fully oxidized $MoO_3$ has a relatively large band gap of 3.1 eV compared to other two materials, with two types of possible charge-transfer

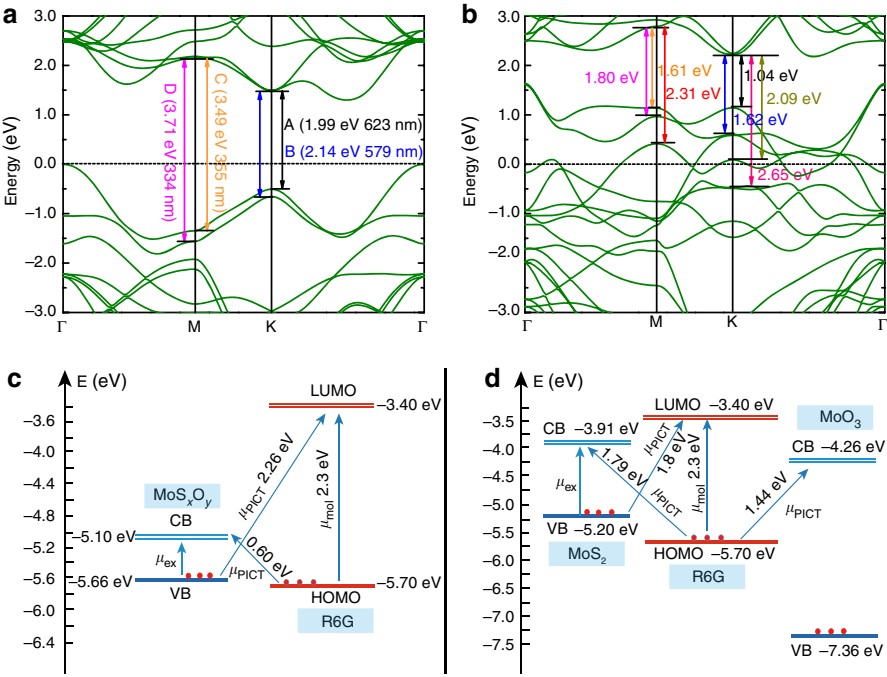

**Fig. 6** Energy-level diagrams illustrating the electronic transitions. The calculated band structures of $MoS_2$ **a** and $MoS_xO_y$ **b** taking Fermi level as reference. Schematic energy-level diagrams of R6G on **c** $MoS_xO_y$ and **d** $MoS_2$ and $MoO_3$ with respect to the vacuum level

transitions from VB to LUMO at 3.96 eV and from HOMO to CB at 1.44 eV. Obviously, neither of charge-transfer transition energies is at or near the excitation laser energy ($\lambda_L = 2.33$ eV), thus leading to low SERS enhancement. In contrast, both pristine $MoS_2$ and partially oxidized $MoS_2$ have much smaller bandgaps of 1.29 and 0.56 eV, respectively, but the CB and VB positions of partially oxidized $MoS_2$ are remarkably downshifted compared to that of pristine $MoS_2$ (Fig. 6c, d), which then causes quite different charge-transfer transitions in two samples. For pristine $MoS_2$, charge-transfer transitions from VB to LUMO and from HOMO to CB occur at 1.80 and 1.79 eV, respectively, whereas the corresponding charge-transfer transitions for partially oxidized $MoS_2$ occur at 2.26 and 0.60 eV, respectively. Although the charge-transfer transitions from VB to LUMO of both materials are possible to induce a charge-transfer resonance ($\lambda_{CT} \approx \lambda_L$), the downshifted VB position after oxygen incorporation in partially oxidized $MoS_2$ makes its charge-transfer transition energy (2.26 eV) much closer to the excitation laser energy when compared with that for pristine $MoS_2$ (1.80 eV), thus a stronger charge-transfer resonance can be expected for partially oxidized $MoS_2$. Moreover, similar to the observation for amorphous ZnO[16], the formation of large quantities of highly localized dangling bonds upon partial oxidation such as Mo–S–O and S–O bonds in our samples can also weaken the constraint to the surface electrons by a redistribution of the electron density in $MoS_2$, which effectively improves the charge-transfer efficiency and further contributes to the SERS enhancement.

Although crucial to the high Raman enhancement on partially oxidized $MoS_2$ is the existence of a charge-transfer resonance, other resonances in the system also contribute to the enhancement. Exciton resonance, which depends on the electronic structure of semiconductors, is recently found to be likely to play a large role in semiconductor SERS[40,42]. In pristine $MoS_2$, these are well-known A, B, C, and D exciton bands located at around 670 nm (1.8 eV), 620 nm (2.00 eV), 365 nm (3.4 eV), and 330 nm (3.71 eV)[43], which are also observed in our ultraviolet-visible spectrum of $MoS_2$ (Supplementary Fig. 12). The A and B peaks are the first members of two excitonic Rydberg series corresponding to the transitions $K4 \rightarrow K5$ and $K1 \rightarrow K5$ at the $K$ point in Fig. 6a, while the C and D peaks are associated with the direct transitions from deep levels in VB to CB at the $M$ point in the Brillouin zone[43]. Interestingly, in partially oxidized $MoS_2$, there are much more direct excitonic transitions to be allowed, which should also be considered as possible contributors to the SERS enhancement. As shown in Fig. 6b, there are four direct transitions with the energies of 1.04, 1.62, 2.09, and 2.65 eV at the $K$ point, and another three direct transitions with the energies of 1.61, 1.80, and 2.31 eV at the $M$ point. This series of allowed direct transitions with a wide range of energy distributions may be responsible for the observed featureless absorption spectrum of partially oxidized $MoS_2$ (Supplementary Fig. 12), which may be a signature of the metallic nature of partially oxidized $MoS_2$ as often being observed in highly doped semiconductor compounds[44]. Based on the above analysis of direct excitonic transitions, only two allowed direct transitions at the $K$ point are in the vicinity of the laser excitation to get effective resonances for pristine $MoS_2$ (Fig. 6a). In comparison, besides $K$-point transitions, another three $M$-point transitions can also participate in resonance with incident laser for partially oxidized $MoS_2$ (Fig. 6b), in which the increased population of exciton resonances may be an important contribution to the charge-transfer effects through vibronic coupling. In addition, note that the molecular transition between the HOMO and LUMO levels of R6G at 2.3 eV is also near the laser (in this case 532.8 nm or 2.33 eV), which provides another resonant pathway to further enhance SERS effect.

Significantly, it should be emphasized that the three resonances involved in semiconductor SERS don't work independently. Instead, the mechanism for Raman enhancement involves the coupling of the charge-transfer resonance with one of the other, more intensely allowed transitions in the molecule-semiconductor system, either molecular resonance or exciton resonance[40]. On coupling, the normally weak charge-transfer resonance borrows intensity from the stronger nearby resonances, which can be expressed by a Herzberg–Teller coupling term $h_{CK}$ and $h_{IV}$ for intensity borrowing from molecular and exciton transitions, respectively. (Supplementary Note 2) Thus, we can now deduce that the high SERS enhancement in partially oxidized $MoS_2$ stems from the coupling of several resonances; namely, charge transfer, molecular, and exciton resonances can all play a part.

Very interestingly, we believe that oxygen extraction in semiconductor oxides may also share the same SERS enhancement mechanism with this oxygen addition in $MoS_2$, in view of creating charge-transfer routes in resonance with incident photons followed by intensity borrowing from nearby resonances. When oxygen is extracted from the lattice of metal oxide, the presence of oxygen vacancies would distort crystal lattice, redistribute electron density, and generate new defect levels ($V_O$) deep in the forbidden band, which, as a result, introduces additional charge-transfer routes between molecular and substrate. Moreover, new exciton resonances would also be introduced, either starting or ending with the $V_O$, which may borrow intensities to nearby charge-transfer resonances through vibronic coupling. Therefore, oxygen vacancies can also bring about remarkable SERS enhancement, which may prevalently exist in semiconductor oxides as SERS substrates through defect engineering.

Based on the studies about the effect of oxygen incorporation (this work) and oxygen extraction (our previous work) on SERS, the observed SERS enhancement arising from oxygen incorporation and oxygen extraction seems to share a unified mechanism that involves: first, additional energy levels facilitate the possibility of charge transfer between semiconductor and analyte molecule, which is in resonance with incident photons. Second, the improvement of exciton resonances brings about stronger intensity borrowing to the charge-transfer resonance in the semiconductor–molecule system. Therefore, by manipulating oxygen atoms in the lattice of semiconductor substrate through either incorporation or extraction to adjust the energy levels of the substrate, the location of both charge-transfer transition and exciton transition would be modulated, which is important to the achievement of highly enhanced SERS signals for specific molecules.

## Discussion

In summary, we have demonstrated that oxygen incorporation is very effective in improving the SERS performance of non-metal-oxide semiconductors, which increases SERS signals by 100,000-fold. The SERS enhancements given by oxygen incorporation continuously increases to a maximum value with increasing oxygen incorporation concentrations as long as its phase structure remains undisrupted; however, it quickly drops to a very small value when the phase change is occurring. A significant point is that this approach can also be easily extended to many other non-metal-oxide semiconductors, such as $WS_2$ and $MoSe_2$. The unique oxygen incorporation-assisted approach not only provides new insights into the CM process in SERS between semiconductor substrates and probe molecules but also may pave the way for the wide application of semiconductor-based SERS.

## Methods

**Synthesis of oxygen-substituted MoS$_2$.** A unit of 1 mmol (NH$_4$)$_6$Mo$_7$O$_{24}$·4H$_2$O and 30 mmol thiourea were dissolved in 40 mL distilled water under vigorous stirring for 30 min to form a homogeneous solution. Then the above solution was transferred into a 100 mL Teflon-lined stainless-steel autoclave, treated at 200 °C for 20 h, and cooled down naturally. The black precipitates formed at the end of the reaction were collected by centrifugation, washed with deionized water and ethanol several times, and finally dried under vacuum at 60 °C for 12 h.

**Synthesis of oxidized MoS$_2$.** For partially oxidized MoS$_2$, the preparation was achieved through controllably annealing of MoS$_2$ samples at the different temperatures and times in air. Different annealing times were used from 10 min to 3 h, while four different annealing temperatures (250, 300, 350, and 400 °C) were designed. A ramp rate of 10 °C min$^{-1}$ was used for the temperature adjustment. Finally, the oxidized MoS$_2$ samples with various degrees of oxidation were obtained.

**Raman measurement.** Raman spectra of R6G molecule deposited on oxygen-incorporated MoS$_2$ samples as substrates were obtained under laser excitation at 532.8 nm. Specifically, R6G aqueous solutions with concentration varied from 10$^{-4}$ to 10$^{-7}$ M were obtained from a stock solution of 10$^{-3}$ M by successive dilution. Then 1 mL of R6G solution with given concentration was combined with 500 μL of oxygen-incorporated MoS$_2$ aqueous dispersion (0.5 mg mL$^{-1}$) followed by 2 h storage in dark to reach the adsorption equilibrium. At last, 20 μL suspension was extracted and dropped onto a cleaned silicon wafer before drying at 60 °C for at least 2 h.

Raman spectra were subsequently acquired on a high-resolution confocal Raman spectrometer (LabRAM HR-800). The spectra were collected by using a 50 × L objective lens for 15 s with a laser spot diameter of about 1 μm and power of 0.3 mW in all acquisitions. Raman spectra from different locations were collected for each sample, with the signal intensity averaged for final analysis to estimate the relative standard deviation values for EFs.

**Data availability.** The relevant data are available within the article and its Supplementary Information files or from the corresponding authors upon reasonable request.

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

## Acknowledgements

This work was supported by the National Natural Science Foundation of China (51372266, 51572286, 21503266, 51772319, and 51772320) and the Outstanding Youth Fund of Jiangsu Province (BK20160011). W.B.L. would like to acknowledge the support from the National Key Research and Development Program of China (2016YFA0203301). F.X.G. acknowledges the support from the National Natural Science Foundation of China (51402204 and 5172200211), the Thousand Young Talents Program, and the Jiangsu Specially-Appointed Professor Program.

## Author contributions

Z.Z.Z. conceived the project and designed the experiments. Z.H.Z., W.B.G., J.N.X., and G.H.L. performed material synthesis, structural characterization, simulation, and Raman measurements. Z.H.Z., S.C., F.X.G., and Z.Z.Z. analysed the data. S.C. and Z.Z.Z. co-wrote the paper. All authors discussed the results and commented on the manuscript.

## Additional information

**Competing interests:** The authors declare that they have no competing financial interests.

