## [Peer Review File · Nature Communications]

Reviewers' Comments:

Reviewer #1 (Remarks to the Author):

In this article the authors present a new way of preparation of a semiconductor substrate for surface-enhanced Raman spectroscopy (SERS), which involves partial oxidation of an MoS₂ surface. Enhancements of up to 1.25×10^5 are obtained with Rhodamine 6G (R6G), and remarkably low limits of detection are also observed. SERS on metal substrates has been found to support enhancement factors of over 10¹¹, allowing single molecule detection. In semiconductors, due to the lack of a plasmon resonance, enhancements have been considerably smaller, at best on the order of 10⁵ to 10⁶. However, even that is often adequate for construction of sensors. Semiconductors have other advantages such as stability, and reproducibility, which in many cases make them superior to metals for certain applications, and are therefore currently a hot topic of research. The SERS enhancement in semiconductors relies on a charge-transfer mechanism rather than plasmon resonances, and the authors correctly identify this as the most important contribution to their remarkable observations. The experimental part of this paper is beautiful and of considerable interest in that it paves the way to construct semiconductor SERS substrates which are sensitive, stable and reproducible. Another important aspect of this work is the ability, by controlling the extent of oxidation, to adjust the energy levels of the substrate to allow specific molecules to be selected by means of adjusting the location of a charge-transfer transition. This aspect deserves more attention by the authors.

The major weakness of this article is the authors' total misunderstanding of the theory of SERS in semiconductors, and therefore needs substantial revision. Firstly, the authors should be aware that there is considerable evidence for charge-transfer through vibronic coupling in their own spectra. Note for example, in figure 4, the lines at 612 cm⁻¹ and 773 cm⁻¹ can be seen to be the most enhanced lines in the spectra. These lines are well-known to be vibronically coupled (J. Phys. Chem. 1984, 88, 5935–5944), and therefore tend to be highly enhanced in SERS wherever charge-transfer is important. Vibronic coupling is very important to Raman enhancement due to proximity to a surface, and is intimately tied up with charge-transfer effects. The mechanism for enhancement (J. Phys. Chem. C 2014, 118, 11120) involves the coupling of the charge-transfer (CT) resonance with one of the other, more intensely allowed transitions in the molecule-semiconductor system, either molecular resonances or exciton resonances. This coupling is said to involve intensity “borrowing”, in which the normally weak CT resonance borrows intensity from the stronger nearby resonances. I think this is what the authors are alluding to when they invoke “dipole-dipole” coupling, but their discussion of it is somewhat vague and disjointed. The way it is presented in this article sounds quite speculative. First note that in order to explain such

a large enhancement, the charge-transfer energy should be near the laser (in this case 532.8 nm or 2.32 eV). As illustrated in their figure 6, it is nowhere near that (0.72 eV in 6c and 1.44 eV in 6d). It is, however, near the transition from the VB of MoS₂ (-5.90 eV) and the LUMO of R6G (-3.40 eV) which lies about 2.5 eV. Notice this is also close to the molecular transition at 2.3 eV (5.70-3.40). Thus, the coupling between the molecular transition and the CT transition should be strong and make a rather large contribution to the enhancement.

I suspect the reason for the considerably higher enhancement in the oxidized substrates is due to differences in the contributions of the excitons, as well as changes in the band structure due to oxidation. These possible contributions to the enhancement from the exciton transitions in the substrate should be explored experimentally. However, the authors seem a bit confused as to how this comes about. This stems partly from their misuse of the calculated band structures indicated in figure 6a and b and S2). The exciton transitions they suggest (0.92 eV and 0.48 eV) are indirect bandgap transitions (figure S2), which are forbidden by the selection rule ($\Delta k = 0$), and therefore cannot contribute to the SERS enhancement. The authors should obtain absorption (UV-vis) spectra of the oxidized substrates (perhaps as a function of % oxygen), and use the band structure calculation and selection rules to assign the allowed exciton spectra. In pure MoS₂ these are well-known (J. Appl. Phys. 81 (12), 1997) A, B, and C excitons are at 670 nm (1.8 eV), 620 nm (2.00 eV) and 365 nm (3.4 eV). In the oxidized substrates, I suggest, at least in figure 6b, the M-point transition ($\Delta k = 0$), which I take to be from about 0.5 eV (VB edge) to about 2.0 eV (CB edge)—just looking at their diagram. There is also another allowed transition from a lower band at about 0.0 eV on the diagram to the one at 2.0 (CB edge), which should lie about 2.0 eV, just in the range of the laser induced CT band and close enough to lend intensity to it. There are also possible excitonic transitions allowed near the K-point, which should also be considered as possible contributors to the SERS enhancement.

In any case, this article is potentially ground-breaking in its construction of a high-enhancement semiconductor for SERS. However, it is weak in its analysis of the mechanism of the enhancement and can be strengthened considerably by proper application of theory.

Reviewer #2 (Remarks to the Author):

Comment:

This is an interesting paper, in which author clearly demonstrated the oxygen-incorporation-assisted approach in improving the SERS performance of non-metal-oxide semiconductors. The experimental results are convinced, which also agree well with the simulations.

In the previous study from this group (S. Cong, et al., Nature Comm. 2015, 6, 7800), the author have found that introducing the oxygen vacancies into semiconductor nanomaterials can effectively enhance the SERS effect. In this paper, the author showed its inverse process that

oxygen incorporation could also effectively magnify the SERS signals of semiconductor materials.

To obtain a general mechanism of those results, the author believe that both effects of oxygen incorporation and oxygen extraction can cause different degrees of lattice distortion, leading to increased electron-transition probability and symmetry-related perturbation.

This is indeed the mechanism suggested by the authors, who suggest the lattice distortion can lead to the remarkable SERS activity of semiconductor nanomaterials. For a very recent study (Angewandte Chemie DOI: 10.1002/ange.201705187), it has been found that the amorphous semiconductor nanomaterials could effectively improve the charge transfer efficiency and magnify the molecular polarization. In particular, the absence of long-range order in the atomic positions can create dangling bonds and band tails, and their arbitrary arrangement makes the energy of the system at the metastable state. The metastable electronic states in amorphous materials, such as the localized band tail states and unpaired electrons in a hybrid orbital, could effectively facilitate the electron escape and transfer. The author may take some references of it. Therefore, this is a very interesting article and is worthy of being published after minor revision.

Reviewer #3 (Remarks to the Author):

I cannot recommend the publication of this manuscript in Nature Communication mainly because the originality and novelty of the manuscript are insufficient. To be accepted in Nature Communication much higher originality and novelty are needed. Semiconductor SERS spectroscopy is not always new. Already many papers and reviews on it have been published. Compared with those previous works the present work may increase enhancement factor of semiconductor SERS but it is not so remarkable to justify the publication in Nature Communication. The theoretical discuss described is also just modified one not original one.

This manuscript does contain interesting results and discussion and thus, it may be publishable but not in Nature Communication but in a more local journal, for example a physical chemistry journal.

The authors did not cite many important papers on semiconductor SERS from Prof. Bing Zhao group of Jilin University, China who is one of the leaders in this field. The authors should consider more fair citation.

Reviewer #4 (Remarks to the Author):

This paper reports an exciting discovery, the kind of result that makes me want to try it out immediately myself (I will of course wait till publication!). First let me say that the paper is well presented, with clear and useful figures and, for the most part, very good and clear English, though some editorial attention is still required. The scientific content is comprehensive and thorough, with very few points where I wanted to ask for more information.

Importantly, the synthesis details (taking into account the SI) appear adequate for others to reproduce the work; this paper is certain to provoke attempts to reproduce its results and so this is essential. The detailed discussion of XPS spectra is useful as a very good way for others to compare their product materials.

The discussion of the sites occupied by oxygen in the lattice is very interesting and credible - there are likely to be a lot of reports on ordering in alloy TMDs and there have already been some (mostly computational) discussions of related effects (for example, the ordering of vacancies). As ternary alloys attract more attention, the possibility of inducing ordering and of exercising control over their structure is very attractive - it will impact on many material properties besides SERS enhancements.

The charge transfer mechanism probably needs more work to prove this, but the comments here are valid and of course the paper would not be complete without attempting some understanding of the mechanism.

I would like to ask the authors if they can make some comments and provide any data on the following two scientific points:

Firstly, the SERS community are always concerned with stability and reproducibility of SERS substrates; these issues have been the major obstacles to commercialisation of many promising ideas. Of course I am not saying that the authors should produce a recipe for a commercially-viable SERS substrate here, but can they tell us anything already about the working lifetime and sensitivity to contamination of their material?

Secondly, I would be very interested to see (perhaps in the SI) the Raman spectra of the different materials without R6G, for two reasons: (i) this will be of assistance to others in reproducing the starting materials, and (ii) I'd expect it to be another indicator of the different types of incorporation of oxygen that the authors present. I was surprised these spectra were not presented. In this context, I note that there is a very strong peak in the Raman spectra at ~ 600 cm^{-1} . This does more or less correspond to an R6G peak, though not a strong one. However, it is where I might also expect local vibrational modes of Mo-O to appear. Can the authors confirm explicitly that **all** the Raman peaks they show are due to R6G? I think it is implied but not

said.

Other minor corrections:

Abstract - should spell out "LOD" in full

p5 lines 110-111

"The significant difference between substitution and oxidation is Mo(VI) can be observed in partial oxidation process but not in oxygen substitution process"

I misunderstood this sentence at first; I think it might help to say "is that Mo(VI) can be observed in the partial oxidation process but not in the oxygen substitution process" - authors please confirm!

p6 line 114

"hydrothermal treatment of $(\text{NH}_4)_6\text{Mo}_7\text{O}_{24} \cdot 4\text{H}_2\text{O}$ is performed"

Does this mean "treatment of thiourea in (NH_4) ..."? I think this is important enough to be absolutely clear in the main text even if it's in the SI. Same comment applies to the figure 1 caption.

p14 line 303

"exhilaratingly"

This is perhaps over the top! I think it's the first time in my career that I've seen this word in a scientific paper so, whilst I appreciated the authors' attempt to use interesting language, I'd tone it down a bit.

p19 line 388

"through Fermi's golden rule"

Not sure why it's necessary to mention FGR here - it's such a general principle that it doesn't add any information to say this.

Answer to reviewer's comments:

We would like to thank all the reviewers for their deep and thorough reviewing of our manuscript. In view of these constructive and helpful comments, we have carefully and substantially revised our manuscript. Here are the detailed responses to the comments of the reviewers.

Replies on comments of reviewer 1:

Comment 1:

In this article, the authors present a new way of preparation of a semiconductor substrate for surface-enhanced Raman spectroscopy (SERS), which involves partial oxidation of an MoS₂ surface. Enhancements of up to 1.25×10^5 are obtained with Rhodamine 6G (R6G), and remarkably low limits of detection are also observed. SERS on metal substrates has been found to support enhancement factors of over 10^{11} , allowing single molecule detection. In semiconductors, due to the lack of a plasmon resonance, enhancements have been considerably smaller, at best on the order of 10^5 to 10^6 . However, even that is often adequate for construction of sensors. Semiconductors have other advantages such as stability, and reproducibility, which in many cases make them superior to metals for certain applications, and are therefore currently a hot topic of research. The SERS enhancement in semiconductors relies on a charge-transfer mechanism rather than plasmon resonances, and the authors correctly identify this as the most important contribution to their remarkable observations. The experimental part of this paper is beautiful and of considerable interest in that it paves the way to construct semiconductor SERS substrates which are sensitive, stable and reproducible

Author reply: Thanks for the referee's comments.

Comment 2:

Another important aspect of this work is the ability, by controlling the extent of oxidation, to adjust the energy levels of the substrate to allow specific molecules to be

selected by means of adjusting the location of a charge-transfer transition. This aspect deserves more attention by the authors. The major weakness of this article is the authors' total misunderstanding of the theory of SERS in semiconductors, and therefore needs substantial revision.

Author reply: We greatly appreciate the valuable suggestion on the understanding of the theory behind semiconductor SERS! We have re-examined our experimental and simulation results very carefully. For example, to obtain more accurate simulation results, we apply the GW method with more computational cost than standard DFT to correct the band energies. Based on these efforts, we have rewritten the discussion section of mechanism studies in the manuscript, especially under the guidance of the reviewer, which emphasizes the importance of the coupling of charge transfer, and molecular and exciton resonances in our system. The revised words are marked in blue in the manuscript, and listed as below:

The mechanism of oxygen-incorporation-assisted SERS enhancement. In our recent studies, it has been established that oxygen vacancies play an important role in enhancing semiconductor SERS effect (Nat. Commun. 2015, 6, 7800). The present findings unexpectedly show that the inverse process of making oxygen vacancies, oxygen incorporation, could also effectively magnify the SERS signals of semiconductor materials. Taken together, it is interesting to see whether the SERS enhancement effect induced by oxygen vacancies and oxygen incorporation reply on the same mechanism. Taking its cue from Lombardi et al.' pioneering theory on the SERS mechanism related to semiconductor materials (J. Phys. Chem. C 2014, 118, 11120-11130), we consider if some resonances such as charge-transfer, exciton and molecular resonances are involved in the mechanism.

Charge-transfer resonance is a resonance Raman-like process associated with the photon-induced charge transfer from the semiconductor band edges to the affinity levels of the adsorbed molecule. This results in a change of the polarizability of the molecule, and consequently amplifies the intensity of its Raman signal (Adv. Mater. 2017, 29, 1604797). For our partially-oxidized MoS₂ samples, there are considerable experimental evidences for charge-transfer through vibronic coupling. Note for

example, in Fig. 4, the lines at 612 cm^{-1} and 773 cm^{-1} (corresponding to in-plane and out-of-plane bending motion of the hydrogen atoms of the xanthen skeleton, respectively) can be seen to be the most enhanced lines in the spectra. These lines are well-known to be vibronically coupled (J. Phys. Chem. 1984, 88, 5935–5944), and therefore tend to be highly enhanced in SERS wherever charge-transfer is important. Comparative analysis of the energy level structures of pristine MoS_2 , partially-oxidized MoS_2 , fully-oxidized MoO_3 and R6G further indicates that partially-oxidized MoS_2 provides significant advantages over other samples in charge transfer. As depicted in Fig. 6, when R6G is used as the target molecule, its HOMO and LUMO levels are at -5.7 eV and -3.4 eV , respectively. Examining the energy levels of the above three semiconductors, we find that the fully-oxidized MoO_3 has a relatively large band gap of 3.1 eV compared to other two materials, with two types of possible charge-transfer transitions from VB to LUMO at 3.96 eV and from HOMO to CB at 1.44 eV . Obviously, neither of charge-transfer transition energies is at or near the excitation laser energy ($\lambda_L=2.33\text{ eV}$), thus leading to low SERS enhancement. In contrast, both pristine MoS_2 and partially-oxidized MoS_2 have much smaller bandgaps of 1.29 and 0.56 eV , respectively, but the CB and VB positions of partially-oxidized MoS_2 is remarkably downshifted compared to that of pristine MoS_2 (Fig. 6c, d), which then causes quite different charge-transfer transitions in two samples. For pristine MoS_2 , charge-transfer transitions from VB to LUMO and from HOMO to CB occur at 1.80 and 1.79 eV , respectively, whereas the corresponding charge-transfer transitions for partially-oxidized MoS_2 occur at 2.26 and 0.60 eV , respectively. Although the charge-transfer transitions from VB to LUMO of both materials are possible to induce a charge-transfer resonance ($\lambda_{CT}\approx\lambda_L$), the downshifted VB position after oxygen incorporation in partially-oxidized MoS_2 makes its charge-transfer transition energy (2.26 eV) much closer to the excitation laser energy when compared with that for pristine MoS_2 (1.80 eV), thus a stronger charge-transfer resonance can be expected for partially-oxidized MoS_2 . Moreover, similar to the observation for amorphous ZnO (Angew. Chem. Int. Ed. 2017, 56, 9851-9855), the formation of large quantities of highly-localized dangling bonds upon partial

oxidation such as Mo-S-O, S-O bonds in our samples can also weaken the constraint to the surface electrons by a redistribution of the electron density in MoS₂, which effectively improve the charge-transfer efficiency and further contribute to the SERS enhancement.

Although crucial to the high Raman enhancement on partially-oxidized MoS₂ is the existence of a charge-transfer resonance, other resonances in the system also contribute to the enhancement. Exciton resonance, which depends on the electronic structure of semiconductors, is recently found to be likely to play a large role in semiconductor SERS (ACS Photonics 2016, 3, 1164-1169; J. Phys. Chem. C 2014, 118, 11120-1113). In pristine MoS₂, these are well-known A, B, C and D exciton bands located at around 670 nm (1.8 eV), 620 nm (2.00 eV), 365 nm (3.4 eV) and 330 nm (3.71 eV) (J. Appl. Phys. 81 (12), 1997), which is also observed in our UV-Vis spectrum of MoS₂ (Supplementary Fig. 12). The A and B peaks are the first members of two excitonic Rydberg series corresponding to the transitions $K4 \rightarrow K5$ and $K1 \rightarrow K5$ at the K point in Fig. 6a, while the C, D peaks are associated with the direct transitions from deep levels in VB to CB at the M point in the Brillouin zone (J. Appl. Phys. 81 (12), 1997). Interestingly, in partially-oxidized MoS₂, there are much more direct excitonic transitions to be allowed, which should also be considered as possible contributors to the SERS enhancement. As shown in Fig. 6b, there are four direct transitions with the energies of 1.04, 1.62, 2.09, 2.65 eV at the K point, and another three direct transitions with the energies of 1.61, 1.80, 2.31 eV at the M point. This series of allowed direct transitions with a wide range of energy distributions may be responsible for the observed featureless absorption spectrum of partially-oxidized MoS₂ (Supplementary Fig. 12), which may be a signature of the metallic nature of partially-oxidized MoS₂ as often being observed in highly-doped semiconductor compounds (Adv. Mater. 2015, 27, 3152–3158). Based on the above analysis of direct excitonic transitions, only two allowed direct transitions at the K point are in the vicinity of the laser excitation to get effective resonances for pristine MoS₂ (Fig. 6a). In comparison, besides K -point transitions, another three M -point transitions can also participate in resonance with incident laser for partially-oxidized MoS₂ (Fig. 6b), in

which the increased population of exciton resonances may be an important contribution to the charge-transfer effects through vibronic coupling. In addition, note that the molecular transition between the HOMO and LUMO levels of R6G at 2.3 eV is also near the laser (in this case 532.8 nm or 2.33 eV), which provides another resonant pathway to further enhance SERS effect.

Significantly, it should be emphasized that the three resonances involved in semiconductor SERS don't work independently. Instead, the mechanism for Raman enhancement involves the coupling of the charge-transfer resonance with one of the other, more intensely allowed transitions in the molecule-semiconductor system, either molecular resonance or exciton resonance (J. Phys. Chem. C 2014, 118, 11120-1113). On coupling, the normally weak charge-transfer resonance borrows intensity from the stronger nearby resonances, which can be expressed by a Herzberg-Teller coupling term h_{CK} and h_{IV} for intensity borrowing from molecular and exciton transitions, respectively. (Supplementary Methods 6) Thus, we can now deduce that the high SERS enhancement in partially-oxidized MoS₂ stems from the coupling of several resonances; namely, charge-transfer, molecular and exciton resonances can all play a part.

Supplementary Figure 12 | UV-Vis absorption spectra of unincorporated MoS₂, oxygen-incorporated MoS₂ and MoO₃.

Figure 6 | Energy level diagrams illustrating the electronic transitions. (a, b) The calculated band structures of MoS₂ and MoS_xO_y taking Fermi level as reference. Schematic energy level diagrams of R6G on (c) MoS_xO_y (d) MoS₂ and MoO₃ with respect to the vacuum level.

6. Herzberg–Teller coupling term (J. Phys. Chem. C 2014, 118, 11120-1113)

A-Terms. Molecule to Semiconductor Charge Transfer

$$R_{ICK}(\omega) = \frac{\mu_{KI} \mu_{IC} h \langle i | Q_K | f \rangle}{((\epsilon_1(\omega) + 2\epsilon_0)^2 + \epsilon_2^2(\omega))((\omega_{IC}^2 - \omega^2) + \gamma_{IC}^2)((\omega_{KI}^2 - \omega^2) + \gamma_{KI}^2)} \quad (6a)$$

$$R_{ICV}(\omega) = \frac{\mu_{VC} \mu_{IC} h \langle i | Q_K | f \rangle}{((\epsilon_1(\omega) + 2\epsilon_0)^2 + \epsilon_2^2(\omega))((\omega_{IC}^2 - \omega^2) + \gamma_{IC}^2)((\omega_{VC}^2 - \omega^2) + \gamma_{VC}^2)} \quad (6b)$$

B-Terms. Semiconductor to Molecule Charge Transfer

$$R_{IVK}(\omega) = \frac{\mu_{VK}\mu_{KI}h_{IV}\langle i|Q_K|f\rangle}{((\epsilon_1(\omega) + 2\epsilon_0)^2 + \epsilon_2^2(\omega))((\omega_{VK}^2 - \omega^2) + \gamma_{VK}^2)((\omega_{KI}^2 - \omega^2) + \gamma_{KI}^2)} \quad (6c)$$

$$R_{KVC}(\omega) = \frac{\mu_{CV}\mu_{VK}h_{KC}\langle i|Q_K|f\rangle}{((\epsilon_1(\omega) + 2\epsilon_0)^2 + \epsilon_2^2(\omega))((\omega_{VK}^2 - \omega^2) + \gamma_{VK}^2)((\omega_{CV}^2 - \omega^2) + \gamma_{CV}^2)} \quad (6d)$$

Very interestingly, we believe that oxygen extraction in semiconductor oxides may also share the same SERS enhancement mechanism with this oxygen addition in MoS₂, in view of creating charge-transfer routes in resonance with incident photons followed by intensity borrowing from nearby resonances. When oxygen is extracted from the lattice of metal oxide, the presence of oxygen vacancies would distort crystal lattice, redistribute electron density, and generate new defect levels (V_o) deep in the forbidden band, which, as a result, introduce additional charge-transfer routes between molecular and substrate. Moreover, new exciton resonances would also be introduced, either starting or ending with the V_o, which may borrow intensities to nearby charge-transfer resonances through vibronic coupling. Therefore, oxygen vacancies can also bring about remarkable SERS enhancement, which may prevalently exist in semiconductor oxides as SERS substrates through defect engineering.

Based on the studies about the effect of oxygen incorporation (this work) and oxygen extraction (our previous work) on SERS, the observed SERS enhancement arising from oxygen incorporation and oxygen extraction seems to share a unified mechanism that involves: 1) additional energy levels facilitates the possibility of charge-transfer between semiconductor and analyte molecule, which is in resonance with incident photons, 2) the improvement of exciton resonances brings about stronger intensity borrowing to the charge-transfer resonance in the semiconductor–molecule system. Therefore, by manipulating oxygen atoms in the lattice of semiconductor substrate through either incorporation or extraction to adjust the energy levels of the substrate, the location of both charge-transfer transition and exciton transition would be

modulated, which is important to the achievement of highly-enhanced SERS signals for specific molecules.

Comment 3:

Firstly, the authors should be aware that there is considerable evidence for charge-transfer through vibronic coupling in their own spectra. Note for example, in figure 4, the lines at 612 cm^{-1} and 773 cm^{-1} can be seen to be the most enhanced lines in the spectra. These lines are well-known to be vibronically coupled (J. Phys. Chem. 1984, 88, 5935–5944), and therefore tend to be highly enhanced in SERS wherever charge-transfer is important. Vibronic coupling is very important to Raman enhancement due to proximity to a surface, and is intimately tied up with charge-transfer effects

Author reply: Thanks for the referee’s comments. As the most important contribution to our remarkable semiconductor SERS enhancement, charge-transfer resonance has been extensively discussed in the revised manuscript as follows:

“Charge-transfer resonance is a resonance Raman-like process associated with the photon-induced charge transfer from the semiconductor band edges to the affinity levels of the adsorbed molecule. This results in a change of the polarizability of the molecule, and consequently amplifies the intensity of its Raman signal (Adv. Mater. 2017, 29, 1604797). For our partially-oxidized MoS₂ samples, there are considerable experimental evidences for charge-transfer through vibronic coupling. Note for example, in Fig. 4, the lines at 612 cm^{-1} and 773 cm^{-1} (corresponding to in-plane and out-of- plane bending motion of the hydrogen atoms of the xanthen skeleton, respectively) can be seen to be the most enhanced lines in the spectra. These lines are well-known to be vibronically coupled (J. Phys. Chem. 1984, 88, 5935–5944), and therefore tend to be highly enhanced in SERS wherever charge-transfer is important. Comparative analysis of the energy level structures of pristine MoS₂, partially-oxidized MoS₂, fully-oxidized MoO₃ and R6G further indicates that partially-oxidized MoS₂ provides significant advantages over other samples in charge transfer. As depicted in Fig. 6, when R6G is used as the target molecule, its HOMO

and LUMO levels are at -5.7 eV and -3.4 eV, respectively. Examining the energy levels of the above three semiconductors, we find that the fully-oxidized MoO₃ has a relatively large band gap of 3.1 eV compared to other two materials, with two types of possible charge-transfer transitions from VB to LUMO at 3.96 eV and from HOMO to CB at 1.44 eV. Obviously, neither of charge-transfer transition energies is at or near the excitation laser energy ($\lambda_L=2.33$ eV), thus leading to low SERS enhancement. In contrast, both pristine MoS₂ and partially-oxidized MoS₂ have much smaller bandgaps of 1.29 and 0.56 eV, respectively, but the CB and VB positions of partially-oxidized MoS₂ is remarkably downshifted compared to that of pristine MoS₂ (Fig. 6c, d), which then causes quite different charge-transfer transitions in two samples. For pristine MoS₂, charge-transfer transitions from VB to LUMO and from HOMO to CB occur at 1.80 and 1.79 eV, respectively, whereas the corresponding charge-transfer transitions for partially-oxidized MoS₂ occur at 2.26 and 0.60 eV, respectively. Although the charge-transfer transitions from VB to LUMO of both materials are possible to induce a charge-transfer resonance ($\lambda_{CT}\approx\lambda_L$), the downshifted VB position after oxygen incorporation in partially-oxidized MoS₂ makes its charge-transfer transition energy (2.26 eV) much closer to the excitation laser energy when compared with that for pristine MoS₂ (1.80 eV), thus a stronger charge-transfer resonance can be expected for partially-oxidized MoS₂.”

Comment 4:

The mechanism for enhancement (J. Phys. Chem. C 2014, 118, 11120) involves the coupling of the charge-transfer (CT) resonance with one of the other, more intensely allowed transitions in the molecule-semiconductor system, either molecular resonances or exciton resonances. This coupling is said to involve intensity “borrowing”, in which the normally weak CT resonance borrows intensity from the stronger nearby resonances.

Author reply: Thanks for the referee’s comments. As mentioned before, the coupling of several resonances in our system has been extensively discussed in the revised manuscript as follows:

“Significantly, it should be emphasized that the three resonances involved in semiconductor SERS don't work independently. Instead, the mechanism for Raman enhancement involves the coupling of the charge-transfer resonance with one of the other, more intensely allowed transitions in the molecule-semiconductor system, either molecular resonance or exciton resonance (J. Phys. Chem. C 2014, 118, 11120-1113). On coupling, the normally weak charge-transfer resonance borrows intensity from the stronger nearby resonances, which can be expressed by a Herzberg-Teller coupling term h_{CK} and h_{IV} for intensity borrowing from molecular and exciton transitions, respectively. (Supplementary Methods 6) Thus, we can now deduce that the high SERS enhancement in partially-oxidized MoS₂ stems from the coupling of several resonances; namely, charge-transfer, molecular and exciton resonances can all play a part.”

Comment 5:

I think this is what the authors are alluding to when they invoke “dipole-dipole” coupling, but their discussion of it is somewhat vague and disjointed. The way it is presented in this article sounds quite speculative. First note that in order to explain such a large enhancement, the charge-transfer energy should be near the laser (in this case 532.8 nm or 2.32 eV). As illustrated in their figure 6, it is nowhere near that (0.72 eV in 6c and 1.44 eV in 6d). It is, however, near the transition from the VB of MoS₂ (-5.90 eV) and the LUMO of R6G (-3.40 eV) which lies about 2.5 eV. Notice this is also close to the molecular transition at 2.3 eV (5.70-3.40). Thus, the coupling between the molecular transition and the CT transition should be strong and make a rather large contribution to the enhancement.

Author reply: Thanks for the referee’s comments. As mentioned before, the following paraps have been added in the revised manuscript as follows:

“Comparative analysis of the energy level structures of pristine MoS₂, partially-oxidized MoS₂, fully-oxidized MoO₃ and R6G further indicates that partially-oxidized MoS₂ provides significant advantages over other samples in charge transfer. As depicted in Fig. 6, when R6G is used as the target molecule, its HOMO

and LUMO levels are at -5.7 eV and -3.4 eV, respectively. Examining the energy levels of the above three semiconductors, we find that the fully-oxidized MoO₃ has a relatively large band gap of 3.1 eV compared to other two materials, with two types of possible charge-transfer transitions from VB to LUMO at 3.96 eV and from HOMO to CB at 1.44 eV. Obviously, neither of charge-transfer transition energies is at or near the excitation laser energy ($\lambda_L=2.33$ eV), thus leading to low SERS enhancement. In contrast, both pristine MoS₂ and partially-oxidized MoS₂ have much smaller bandgaps of 1.29 and 0.56 eV, respectively, but the CB and VB positions of partially-oxidized MoS₂ is remarkably downshifted compared to that of pristine MoS₂ (Fig. 6c, d), which then causes quite different charge-transfer transitions in two samples. For pristine MoS₂, charge-transfer transitions from VB to LUMO and from HOMO to CB occur at 1.80 and 1.79 eV, respectively, whereas the corresponding charge-transfer transitions for partially-oxidized MoS₂ occur at 2.26 and 0.60 eV, respectively. Although the charge-transfer transitions from VB to LUMO of both materials are possible to induce a charge-transfer resonance ($\lambda_{CT}\approx\lambda_L$), the downshifted VB position after oxygen incorporation in partially-oxidized MoS₂ makes its charge-transfer transition energy (2.26 eV) much closer to the excitation laser energy when compared with that for pristine MoS₂ (1.80 eV), thus a stronger charge-transfer resonance can be expected for partially-oxidized MoS₂.”

“In addition, note that the molecular transition between the HOMO and LUMO levels of R6G at 2.3 eV is also near the laser (in this case 532.8 nm or 2.33 eV), which provides another resonant pathway to further enhance SERS effect.”

Furthermore, according to referee 2's suggestions, we also discuss the role of dangling bond on charge-transfer transition and SERS performance. In a very recent study (Angew. Chem. Int. Ed. 2017, 56, 9851-9855) provide by referee 2, it has been found that the amorphous ZnO could effectively improve the charge transfer efficiency and magnify the molecular polarization. In particular, the absence of long-range order in the atomic positions can create dangling bonds and band tails, and their arbitrary arrangement makes the energy of the system at the metastable state. The metastable electronic states in amorphous materials, such as the localized band tail states and

unpaired electrons in a hybrid orbital, could effectively facilitate the electron escape and transfer. Referee 2 believes that the dangling bonds in our partially-oxidized samples also have a similar effect on SERS with those in amorphous ZnO, so an additional paragraph is added in the revised manuscript as follows:

“Moreover, similar to the observation for amorphous ZnO (*Angew. Chem. Int. Ed.* 2017, 56, 9851-9855), the formation of large quantities of highly-localized dangling bonds upon partial oxidation such as Mo-S-O, S-O bonds in our samples can also weaken the constraint to the surface electrons by a redistribution of the electron density in MoS₂, which effectively improve the charge-transfer efficiency and further contribute to the SERS enhancement.”

Comment 6:

I suspect the reason for the considerably higher enhancement in the oxidized substrates is due to differences in the contributions of the excitons, as well as changes in the band structure due to oxidation. These possible contributions to the enhancement from the exciton transitions in the substrate should be explored experimentally. However, the authors seem a bit confused as to how this comes about. This stems partly from their misuse of the calculated band structures indicated in figure 6a and b and S2). The exciton transitions they suggest (0.92 eV and 0.48 eV) are indirect bandgap transitions (figure S2), which are forbidden by the selection rule ($\Delta k = 0$), and therefore cannot contribute to the SERS enhancement. The authors should obtain absorption (UV-vis) spectra of the oxidized substrates (perhaps as a function of % oxygen), and use the band structure calculation and selection rules to assign the allowed exciton spectra. In pure MoS₂ these are well-known (*J. Appl. Phys.* 81 (12), 1997) A, B, and C excitons are at 670 nm (1.8 eV), 620 nm (2.00 eV) and 365 nm (3.4 eV). In the oxidized substrates, I suggest, at least in figure 6b, the M-point transition ($\Delta k = 0$), which I take to be from about 0.5 eV (VB edge) to about 2.0 eV (CB edge)—just looking at their diagram. There is also another allowed transition from a lower band at about 0.0 eV on the diagram to the one at 2.0 (CB edge), which should lie about 2.0 eV, just in the range of the laser induced CT

band and close enough to lend intensity to it. There are also possible excitonic transitions allowed near the K-point, which should also be considered as possible contributors to the SERS enhancement.

Author reply: Thanks for the referee's comments. As mentioned before, the following paragraphs about exciton resonance have been added in the revised manuscript as follows:

“Although crucial to the high Raman enhancement on partially-oxidized MoS₂ is the existence of a charge-transfer resonance, other resonances in the system also contribute to the enhancement. Exciton resonance, which depends on the electronic structure of semiconductors, is recently found to be likely to play a large role in semiconductor SERS (ACS Photonics 2016, 3, 1164-1169; J. Phys. Chem. C 2014, 118, 11120-1113). In pristine MoS₂, these are well-known A, B, C and D exciton bands located at around 670 nm (1.8 eV), 620 nm (2.00 eV), 365 nm (3.4 eV) and 330 nm (3.71 eV) (J. Appl. Phys. 81 (12), 1997), which is also observed in our UV-Vis spectrum of MoS₂ (Supplementary Fig. 12). The A and B peaks are the first members of two excitonic Rydberg series corresponding to the transitions $K4 \rightarrow K5$ and $K1 \rightarrow K5$ at the *K* point in Fig. 6a, while the C, D peaks are associated with the direct transitions from deep levels in VB to CB at the *M* point in the Brillouin zone (J. Appl. Phys. 81 (12), 1997). Interestingly, in partially-oxidized MoS₂, there are much more direct excitonic transitions to be allowed, which should also be considered as possible contributors to the SERS enhancement. As shown in Figure 6b, there are four direct transitions with the energies of 1.04, 1.62, 2.09, 2.65 eV at the *K* point, and another three direct transitions with the energies of 1.61, 1.80, 2.31 eV at the *M* point. This series of allowed direct transitions with a wide range of energy distributions may be responsible for the observed featureless absorption spectrum of partially-oxidized MoS₂ (Supplementary Fig. 12), which may be a signature of the metallic nature of partially-oxidized MoS₂ as often being observed in highly-doped semiconductor compounds (Adv. Mater. 2015, 27, 3152–3158). Based on the above analysis of direct excitonic transitions, only two allowed direct transitions at the *K* point are in the vicinity of the laser excitation to get effective resonances for pristine MoS₂ (Fig. 6a).

In comparison, besides *K*-point transitions, another three *M*-point transitions can also participate in resonance with incident laser for partially-oxidized MoS₂ (Fig. 6b), in which the increased population of exciton resonances may be an important contribution to the charge-transfer effects through vibronic coupling.”

Comment 7:

In any case, this article is potentially ground-breaking in its construction of a high-enhancement semiconductor for SERS. However, it is weak in its analysis of the mechanism of the enhancement and can be strengthened considerably by proper application of theory.

Author reply: Thanks for the referee’s comments. Under your kind guidance, we believe that the analysis of SERS enhancement mechanism in the revised manuscript has been considerably strengthened on the basis of the coupling of charge transfer, and molecular and exciton resonances.

Replies on comments of reviewer 2:

Comment 1:

This is an interesting paper, in which author clearly demonstrated the oxygen-incorporation-assisted approach in improving the SERS performance of non-metal-oxide semiconductors. The experimental results are convinced, which also agree well with the simulations.

In the previous study from this group (S. Cong, et al., Nature Comm. 2015, 6, 7800), the author have found that introducing the oxygen vacancies into semiconductor nanomaterials can effectively enhance the SERS effect. In this paper, the author showed its inverse process that oxygen incorporation could also effectively magnify the SERS signals of semiconductor materials.

Author reply: Thanks for the referee’s comments.

Comment 2:

To obtain a general mechanism of those results, the author believes that both effects of oxygen incorporation and oxygen extraction can cause different degrees of lattice distortion, leading to increased electron-transition probability and symmetry-related perturbation.

This is indeed the mechanism suggested by the authors, who suggest the lattice distortion can lead to the remarkable SERS activity of semiconductor nanomaterials. For a very recent study (Angewandte Chemie DOI: 10.1002/ange.201705187), it has been found that the amorphous semiconductor nanomaterials could effectively improve the charge transfer efficiency and magnify the molecular polarization. In particular, the absence of long-range order in the atomic positions can create dangling bonds and band tails, and their arbitrary arrangement makes the energy of the system at the metastable state. The metastable electronic states in amorphous materials, such as the localized band tail states and unpaired electrons in a hybrid orbital, could effectively facilitate the electron escape and transfer. The author may take some references of it.

Therefore, this is a very interesting article and is worthy of being published after minor revision.

Author reply: Thank you for this valuable comment and suggestion! Actually, we also believe that the dangling bond in our partially-oxidized samples have a similar effect on SERS with those in amorphous ZnO reported by Guo's group. In our experiment, the formation of large quantities of highly-localized dangling bonds upon partial oxidation such as Mo-S-O, S-O bond can also significantly alter the local electronic properties and induce an enhanced interface interaction between adsorbent and adsorbate (as validated by our calculation results of Bader charges, Supplementary Fig. 4), which effectively facilitate the electron escape and transfer and further contribute to the SERS enhancement. Thus, we have cited these valuable papers from Guo's group, and added several paragraphs in the revised manuscript as follows:

“Further, a recent breakthrough made by Guo et al. has surprisingly transformed

non-SERS or weak Cu₂O, ZnO and MoO₃ substrates into highly SERS-active substrates with EFs as high as 1.8×10^7 and LODs as low as 10^{-9} M by oxygen vacancy engineering, respectively (Analyst. 2017, 142, 326; Adv. Mater. 2017, 29, 1604797; Angew. Chem. Int. Ed. 2017, 56, 9851-9855). Specially, Wang et al. indicates that the remarkable SERS activity of amorphous ZnO nanocages can be attributed to the high-efficiency interfacial charge transfer, which is assisted by the numerous metastable electronic states of amorphous ZnO nanocages (Angew. Chem. Int. Ed. 2017, 56, 9851-9855).”

“In such distorted regions, local electronic properties would be significantly altered from those obtained in the undistorted regions (as validated by our calculation results of Bader charges, Supplementary Fig. 4) (J. Am. Chem. Soc. 2013, 135, 17881-17888), probably impacting on the charge-transfer efficiency and magnifying the molecular polarization, ultimately resulting in enhanced SERS signals for molecules attached to them, just as demonstrated in amorphous ZnO reported by Guo’s group (Angew. Chem. Int. Ed. 2017, 56, 9851-9855).”

“Moreover, similar to the observation for amorphous ZnO (Angew. Chem. Int. Ed. 2017, 56, 9851-9855), the formation of large quantities of highly-localized dangling bonds upon partial oxidation such as Mo-S-O, S-O bonds in our samples can also weaken the constraint to the surface electrons by a redistribution of the electron density in MoS₂, which effectively improve the charge-transfer efficiency and further contribute to the SERS enhancement.”

Replies on comments of reviewer 3:

Comment 1:

I cannot recommend the publication of this manuscript in Nature Communication mainly because the originality and novelty of the manuscript are insufficient. To be accepted in Nature Communication much higher originality and novelty are needed . Semiconductor SERS spectroscopy is not always new. Already many papers

and reviews on it have been published. Compared with those previous works the present work may increase enhancement factor of semiconductor SERS but it is not so remarkable to justify the publication in Nature Communication. The theoretical discuss described is also just modified one not original one.

Author reply: Thanks for the referee's comments!

Firstly, we would like to further emphasize the originality and novelty of our manuscript. Semiconductor materials as SERS substrates has already been widely studied, however, the application of these materials is still seriously impeded by their low SERS enhancement and inferior detection sensitivity, especially for non-metal-oxide semiconductor materials. To the best of our knowledge, there are few viable routes to enormously boosting the SERS signals of non-metal-oxide semiconductor materials. For the first time, we demonstrate that oxygen incorporation in MoS₂ even with trace concentrations can not only dramatically increase enhancement factors by up to 100,000 folds compared with oxygen-unincorporated samples, but also endow MoS₂ with low limit of detection below 10⁻⁷ M. Perhaps the enhancement factors of our materials are not the topmost, but we believe that the unique oxygen-incorporation-assisted approach may provide new insights into the CM process in SERS between semiconductor substrates and probe molecules, and in our recent works we always hammer away at finding a universal methodology for the design and optimization of semiconductors as advanced SERS substrates.

Secondly, in the present work, we try to provide a unified mechanism for both oxygen-incorporation-assisted and oxygen-vacancies-assisted SERS enhancement at a much deeper level. The mechanism involves: 1) additional energy levels facilitates the possibility of charge-transfer between semiconductor and analyte molecule, which is in resonance with incident photons, 2) the improvement of exciton resonances brings about stronger intensity borrowing to the charge-transfer resonance in the semiconductor–molecule system. Therefore, by manipulating oxygen atoms in the lattice of semiconductor substrate through either incorporation or extraction to adjust the energy levels of the substrate, the location of both charge-transfer transition and exciton transition would be modulated, which is important to the achievement of

highly-enhanced SERS signals for specific molecules.

Comment 2:

The authors did not cite many important papers on semiconductor SERS from Prof. Bing Zhao group of Jilin University, China who is one of the leaders in this field. The authors should consider more fair citation.

Author reply: Thanks for the referee's valuable comments and suggestion. Prof. Bing Zhao is a respectable scientist and one of the leaders in semiconductor SERS field. We have cited several important papers on semiconductor SERS from Prof. Bing Zhao group in the revised manuscript.

“Zhao et al. have also reported the ultra-sensitive detection of benzoic acid analogues on TiO₂-based SERS substrates with LODs as low as 10⁻⁸ M, which takes advantage of the promoted charge transfer between the substrate and adsorbed molecules via oxygen vacancies (Phys. Chem. Chem. Phys. 2017, 19, 18731-18738; Phys. Chem. Chem. Phys. 2017, 19, 22302-22308; Phys. Chem. Chem. Phys. 2017, 19, 11212-11219).”

Replies to additional comments:

Comment 1:

The editors have asked me to make additional comments on the advancement and novelty of this work in the light of a previously published article, apparently from the same laboratory (S. Cong, et al., Nature Commun. 2015, 6, 7800). There are several apparent similarities between the two articles. Both involve modification of a semiconductor surface, which shows a relatively weak Raman enhancement. The modification results in a considerable increase in enhancement and therefore is suggested as a ground-breaking improvement in our ability to exploit semiconductors as high quality SERS substrates. The differences between the two articles are that the substrates are quite different (WO₃ and MoS₂). One was modified by reduction, and the other by oxidation. Oddly, their theoretical interpretation of their results was quite

good in the first article, and rather vague and unlikely in the second.

Author reply: Thanks for the referee's comments and suggestions. In this revised manuscript, we have rewritten the discussion section of mechanism studies to make our theoretical interpretation more plausible. Based on our experimental and DFT simulation results in this work, we have systematically analyzed the contributions to an enhanced SERS from several resonances, including charge transfer, molecular and exciton resonances, and emphasize the importance of the coupling of these resonances in our system. Furthermore, at the end of this paper, we have taken a unified view to illustrate the SERS enhancement of semiconductors upon either oxygen incorporation (present work) or oxygen-extraction (our previous work), which is original as theoretical discussions about semiconductor SERS.

The mechanism of oxygen-incorporation-assisted SERS enhancement. In our recent studies, it has been established that oxygen vacancies play an important role in enhancing semiconductor SERS effect (Nat. Commun. 2015, 6, 7800). The present findings unexpectedly show that the inverse process of making oxygen vacancies, oxygen incorporation, could also effectively magnify the SERS signals of semiconductor materials. Taken together, it is interesting to see whether the SERS enhancement effect induced by oxygen vacancies and oxygen incorporation reply on the same mechanism. Taking its cue from Lombardi et al.' pioneering theory on the SERS mechanism related to semiconductor materials (J. Phys. Chem. C 2014, 118, 11120-11130), we consider if some resonances such as charge-transfer, exciton and molecular resonances are involved in the mechanism.

Charge-transfer resonance is a resonance Raman-like process associated with the photon-induced charge transfer from the semiconductor band edges to the affinity levels of the adsorbed molecule. This results in a change of the polarizability of the molecule, and consequently amplifies the intensity of its Raman signal (Adv. Mater. 2017, 29, 1604797). For our partially-oxidized MoS₂ samples, there are considerable experimental evidences for charge-transfer through vibronic coupling. Note for example, in Fig. 4, the lines at 612 cm⁻¹ and 773 cm⁻¹ (corresponding to in-plane and out-of- plane bending motion of the hydrogen atoms of the xanthene

skeleton, respectively) can be seen to be the most enhanced lines in the spectra. These lines are well-known to be vibronically coupled (J. Phys. Chem. 1984, 88, 5935–5944), and therefore tend to be highly enhanced in SERS wherever charge-transfer is important. Comparative analysis of the energy level structures of pristine MoS₂, partially-oxidized MoS₂, fully-oxidized MoO₃ and R6G further indicates that partially-oxidized MoS₂ provides significant advantages over other samples in charge transfer. As depicted in Fig. 6, when R6G is used as the target molecule, its HOMO and LUMO levels are at -5.7 eV and -3.4 eV, respectively. Examining the energy levels of the above three semiconductors, we find that the fully-oxidized MoO₃ has a relatively large band gap of 3.1 eV compared to other two materials, with two types of possible charge-transfer transitions from VB to LUMO at 3.96 eV and from HOMO to CB at 1.44 eV. Obviously, neither of charge-transfer transition energies is at or near the excitation laser energy ($\lambda_L=2.33$ eV), thus leading to low SERS enhancement. In contrast, both pristine MoS₂ and partially-oxidized MoS₂ have much smaller bandgaps of 1.29 and 0.56 eV, respectively, but the CB and VB positions of partially-oxidized MoS₂ is remarkably downshifted compared to that of pristine MoS₂ (Fig. 6c, d), which then causes quite different charge-transfer transitions in two samples. For pristine MoS₂, charge-transfer transitions from VB to LUMO and from HOMO to CB occur at 1.80 and 1.79 eV, respectively, whereas the corresponding charge-transfer transitions for partially-oxidized MoS₂ occur at 2.26 and 0.60 eV, respectively. Although the charge-transfer transitions from VB to LUMO of both materials are possible to induce a charge-transfer resonance ($\lambda_{CT}\approx\lambda_L$), the downshifted VB position after oxygen incorporation in partially-oxidized MoS₂ makes its charge-transfer transition energy (2.26 eV) much closer to the excitation laser energy when compared with that for pristine MoS₂ (1.80 eV), thus a stronger charge-transfer resonance can be expected for partially-oxidized MoS₂. Moreover, similar to the observation for amorphous ZnO (Angew. Chem. Int. Ed. 2017, 56, 9851-9855), the formation of large quantities of highly-localized dangling bonds upon partial oxidation such as Mo-S-O, S-O bonds in our samples can also weaken the constraint to the surface electrons by a redistribution of the electron density in MoS₂, which

effectively improve the charge-transfer efficiency and further contribute to the SERS enhancement.

Although crucial to the high Raman enhancement on partially-oxidized MoS₂ is the existence of a charge-transfer resonance, other resonances in the system also contribute to the enhancement. Exciton resonance, which depends on the electronic structure of semiconductors, is recently found to be likely to play a large role in semiconductor SERS (ACS Photonics 2016, 3, 1164-1169; J. Phys. Chem. C 2014, 118, 11120-1113). In pristine MoS₂, these are well-known A, B, C and D exciton bands located at around 670 nm (1.8 eV), 620 nm (2.00 eV), 365 nm (3.4 eV) and 330 nm (3.71 eV) (J. Appl. Phys. 81 (12), 1997), which is also observed in our UV-Vis spectrum of MoS₂ (Supplementary Fig. 12). The A and B peaks are the first members of two excitonic Rydberg series corresponding to the transitions $K4 \rightarrow K5$ and $K1 \rightarrow K5$ at the K point in Fig. 6a, while the C, D peaks are associated with the direct transitions from deep levels in VB to CB at the M point in the Brillouin zone (J. Appl. Phys. 81 (12), 1997). Interestingly, in partially-oxidized MoS₂, there are much more direct excitonic transitions to be allowed, which should also be considered as possible contributors to the SERS enhancement. As shown in Fig. 6b, there are four direct transitions with the energies of 1.04, 1.62, 2.09, 2.65 eV at the K point, and another three direct transitions with the energies of 1.61, 1.80, 2.31 eV at the M point. This series of allowed direct transitions with a wide range of energy distributions may be responsible for the observed featureless absorption spectrum of partially-oxidized MoS₂ (Supplementary Fig. 12), which may be a signature of the metallic nature of partially-oxidized MoS₂ as often being observed in highly-doped semiconductor compounds (Adv. Mater. 2015, 27, 3152–3158). Based on the above analysis of direct excitonic transitions, only two allowed direct transitions at the K point are in the vicinity of the laser excitation to get effective resonances for pristine MoS₂ (Fig. 6a). In comparison, besides K -point transitions, another three M -point transitions can also participate in resonance with incident laser for partially-oxidized MoS₂ (Fig. 6b), in which the increased population of exciton resonances may be an important contribution to the charge-transfer effects through vibronic coupling. In addition, note

that the molecular transition between the HOMO and LUMO levels of R6G at 2.3 eV is also near the laser (in this case 532.8 nm or 2.33 eV), which provides another resonant pathway to further enhance SERS effect.

Significantly, it should be emphasized that the three resonances involved in semiconductor SERS don't work independently. Instead, the mechanism for Raman enhancement involves the coupling of the charge-transfer resonance with one of the other, more intensely allowed transitions in the molecule-semiconductor system, either molecular resonance or exciton resonance (J. Phys. Chem. C 2014, 118, 11120-1113). On coupling, the normally weak charge-transfer resonance borrows intensity from the stronger nearby resonances, which can be expressed by a Herzberg-Teller coupling term h_{CK} and h_{IV} for intensity borrowing from molecular and exciton transitions, respectively. (Supplementary Methods 6) Thus, we can now deduce that the high SERS enhancement in partially-oxidized MoS₂ stems from the coupling of several resonances; namely, charge-transfer, molecular and exciton resonances can all play a part.

Supplementary Figure 12 | UV-Vis absorption spectra of unincorporated MoS₂, oxygen-incorporated MoS₂ and MoO₃.

Figure 6 | Energy level diagrams illustrating the electronic transitions. (a, b) The calculated band structures of MoS₂ and MoS_xO_y taking Fermi level as reference. Schematic energy level diagrams of R6G on (c) MoS_xO_y (d) MoS₂ and MoO₃ with respect to the vacuum level.

6. Herzberg–Teller coupling term (J. Phys. Chem. C 2014, 118, 11120-1113)

A-Terms. Molecule to Semiconductor Charge Transfer

$$R_{ICK}(\omega) = \frac{\mu_{KI} \mu_{IC} h \langle i | Q_K | f \rangle}{((\epsilon_1(\omega) + 2\epsilon_0)^2 + \epsilon_2^2(\omega))((\omega_{IC}^2 - \omega^2) + \gamma_{IC}^2)((\omega_{KI}^2 - \omega^2) + \gamma_{KI}^2)} \quad (6a)$$

$$R_{ICV}(\omega) = \frac{\mu_{VC} \mu_{IC} h \langle i | Q_K | f \rangle}{((\epsilon_1(\omega) + 2\epsilon_0)^2 + \epsilon_2^2(\omega))((\omega_{IC}^2 - \omega^2) + \gamma_{IC}^2)((\omega_{VC}^2 - \omega^2) + \gamma_{VC}^2)} \quad (6b)$$

B-Terms. Semiconductor to Molecule Charge Transfer

$$R_{IVK}(\omega) = \frac{\mu_{VK}\mu_{KI}h_{IV}\langle i|Q_K|f\rangle}{((\epsilon_1(\omega) + 2\epsilon_0)^2 + \epsilon_2^2(\omega))((\omega_{VK}^2 - \omega^2) + \gamma_{VK}^2)((\omega_{KI}^2 - \omega^2) + \gamma_{KI}^2)} \quad (6c)$$

$$R_{KVC}(\omega) = \frac{\mu_{CV}\mu_{VK}h_{KC}\langle i|Q_K|f\rangle}{((\epsilon_1(\omega) + 2\epsilon_0)^2 + \epsilon_2^2(\omega))((\omega_{VK}^2 - \omega^2) + \gamma_{VK}^2)((\omega_{CV}^2 - \omega^2) + \gamma_{CV}^2)} \quad (6d)$$

Very interestingly, we believe that oxygen extraction in semiconductor oxides may also share the same SERS enhancement mechanism with this oxygen addition in MoS₂, in view of creating charge-transfer routes in resonance with incident photons followed by intensity borrowing from nearby resonances. When oxygen is extracted from the lattice of metal oxide, the presence of oxygen vacancies would distort crystal lattice, redistribute electron density, and generate new defect levels (V_o) deep in the forbidden band, which, as a result, introduce additional charge-transfer routes between molecular and substrate. Moreover, new exciton resonances would also be introduced, either starting or ending with the V_o, which may borrow intensities to nearby charge-transfer resonances through vibronic coupling. Therefore, oxygen vacancies can also bring about remarkable SERS enhancement, which may prevalently exist in semiconductor oxides as SERS substrates through defect engineering.

Based on the studies about the effect of oxygen incorporation (this work) and oxygen extraction (our previous work) on SERS, the observed SERS enhancement arising from oxygen incorporation and oxygen extraction seems to share a unified mechanism that involves: 1) additional energy levels facilitates the possibility of charge-transfer between semiconductor and analyte molecule, which is in resonance with incident photons, 2) the improvement of exciton resonances brings about stronger intensity borrowing to the charge-transfer resonance in the semiconductor–molecule system. Therefore, by manipulating oxygen atoms in the lattice of semiconductor substrate through either incorporation or extraction to adjust the energy levels of the substrate, the location of both charge-transfer transition and exciton transition would be

modulated, which is important to the achievement of highly-enhanced SERS signals for specific molecules.

Comment 2:

In retrospect, now reading the first article, the current article does not seem as novel or ground-breaking as I previously thought. There are differences, as I pointed out above, and they are important, as part of the overall narrative of SERS in semiconductors, and together they represent important and novel advances in these materials. They should definitely be published in some journal, but I am not so convinced as to whether they are of sufficient novelty for Nature Communications.

Author reply: Thanks for the referee's comments. Firstly, we would like to further emphasize the originality and novelty of our manuscript. Semiconductor materials as SERS substrates has already been widely studied, however, the application of these materials is still seriously impeded by their low SERS enhancement and inferior detection sensitivity, especially for non-metal-oxide semiconductor materials. To the best of our knowledge, there are few viable routes to enormously boosting the SERS signals of non-metal-oxide semiconductor materials. For the first time, we demonstrate that oxygen incorporation in MoS₂ even with trace concentrations can not only dramatically increase enhancement factors by up to 100,000 folds compared with oxygen-unincorporated samples, but also endow MoS₂ with low limit of detection below 10⁻⁷ M. Perhaps the enhancement factors of our materials are not the topmost, but we believe that the unique oxygen-incorporation-assisted approach may provide new insights into the CM process in SERS between semiconductor substrates and probe molecules, and in our recent works we always hammer away at finding a universal methodology for the design and optimization of semiconductors as advanced SERS substrates.

Secondly, in the present work, we try to provides a unified mechanism for both oxygen-incorporation-assisted and oxygen- vacancies-assisted SERS enhancement at a much deeper level. The mechanism involves: 1) additional energy levels facilitates the possibility of charge-transfer between semiconductor and analyte molecule, which

is in resonance with incident photons, 2) the improvement of exciton resonances brings about stronger intensity borrowing to the charge-transfer resonance in the semiconductor–molecule system. Therefore, by manipulating oxygen atoms in the lattice of semiconductor substrate through either incorporation or extraction to adjust the energy levels of the substrate, the location of both charge-transfer transition and exciton transition would be modulated, which is important to the achievement of highly-enhanced SERS signals for specific molecules.

Replies on comments of reviewer 4:

Comment 1:

This paper reports an exciting discovery, the kind of result that makes me want to try it out immediately myself (I will of course wait till publication!). First let me say that the paper is well presented, with clear and useful figures and, for the most part, very good and clear English, though some editorial attention is still required. The scientific content is comprehensive and thorough, with very few points where I wanted to ask for more information.

Importantly, the synthesis details (taking into account the SI) appear adequate for others to reproduce the work; this paper is certain to provoke attempts to reproduce its results and so this is essential. The detailed discussion of XPS spectra is useful as a very good way for others to compare their product materials.

The discussion of the sites occupied by oxygen in the lattice is very interesting and credible - there are likely to be a lot of reports on ordering in alloy TMDs and there have already been some (mostly computational) discussions of related effects (for example, the ordering of vacancies). As ternary alloys attract more attention, the possibility of inducing ordering and of exercising control over their structure is very attractive - it will impact on many material properties besides SERS enhancements.

The charge transfer mechanism probably needs more work to prove this, but the comments here are valid and of course the paper would not be complete without attempting some understanding of the mechanism.

Author reply: Thanks for the referee's comments!

Comment 2:

I would like to ask the authors if they can make some comments and provide any data on the following two scientific points:

Firstly, the SERS community are always concerned with stability and reproducibility of SERS substrates; these issues have been the major obstacles to commercialisation of many promising ideas. Of course I am not saying that the authors should produce a recipe for a commercially-viable SERS substrate here, but can they tell us anything already about the working lifetime and sensitivity to contamination of their material?

Author reply: Thanks for the referee's kind comments and suggestion. We have carefully examined the working lifetime as well as sensitivity to contamination of our partially-oxidized samples as SERS substrates. Detailed experimental process and results are added in the SI, and also listed as follows:

Supplementary Figure 14 | Working lifetime of the SERS substrate. SERS performances of the partially-oxidized MoS_2 samples obtained at 300°C for 40 mins

are recorded every 10 minutes (a) and before/after two weeks of storage in the open air (c). Reproducibility and stability are two important issues for SERS substrate performance. Firstly, the relative standard deviation (RSD) of the characteristic peak of R6G at 612 cm^{-1} is used to estimate the reproducibility of the SERS signal. (b) shows the SERS-RSD spectra of R6G molecules, randomly collected from 30 positions of six substrates. The RSD value is calculated to be 11.72%, indicating good reproducibility. Secondly, to investigate this stability of substrates, the partially-oxidized samples are kept in the open air for two weeks and then then performed SERS measurement on them. The collected SERS spectra after two weeks of storage in the open air are compared with those obtained from the freshly prepared substrate (c). It is noted that neither a shift in the major Raman peaks nor a significant change in Raman intensity occurred for two weeks, and the RSD value remains as low as 14.37% (d). The result reveals that the as-prepared substrate is stable for at least two weeks.

Supplementary Figure 15 | Sensitivity to contamination of the SERS substrate.

SERS performances of partially-oxidized samples obtained at 300 °C for 40 mins upon contamination with (a) methyl green (MG) or (c) methylene blue (MB). Another major concern in using SERS substrate is whether there will be interferences caused by the interaction of the contaminants. R6G-MG and R6G-MB solutions are prepared as model systems by mixing 1 mL of the 1×10^{-4} M solution of R6G with 9 mL of the 1×10^{-5} M solution of MG or MB solution. (a) shows the SERS spectra of R6G, MG and R6G-MG mixture, while (c) shows the SERS spectra of R6G, MB and R6G-MB mixture. As evident from these figures, the SERS spectrum of R6G seems not to be inhibited by the presence of MG or MB, showing good selectivity and sensitivity for R6G detection.

Comment 3:

Secondly, I would be very interested to see (perhaps in the SI) the Raman spectra of the different materials without R6G, for two reasons: (i) this will be of assistance to others in reproducing the starting materials, and (ii) I'd expect it to be another indicator of the different types of incorporation of oxygen that the authors present. I was surprised these spectra were not presented. In this context, I note that there is a very strong peak in the Raman spectra at $\sim 600 \text{ cm}^{-1}$. This does more or less correspond to an R6G peak, though not a strong one. However, it is where I might also expect local vibrational modes of Mo-O to appear. Can the authors confirm explicitly that **all** the Raman peaks they show are due to R6G? I think it is implied but not said.

Author reply: Thanks for the referee's valuable suggestion. We have systematically examined the Raman spectra of R6G, pristine MoS₂ without R6G loading and partially-oxidized MoS₂ samples without R6G loading. With an increase of the oxidation temperature, the Raman spectra undergo obvious changes for MoS₂ samples. Specifically, the Raman bands of the samples at low-temperature treatment (MoS₂-200 °C and MoS₂-300 °C) indicate the coexistence of Mo-S and Mo-O bonds, while for MoS₂-350 °C sample after high-temperature treatment, the disappearance of A_{1g} (405 cm⁻¹) band evidences a decreased amount of S component as well as the

phase transition from pristine MoS₂ towards fully-oxidized MoO₃ (a). Furthermore, by comparing the Raman spectra of R6G and partially-oxidized MoS₂ samples without R6G loading (b), it is concluded that the peak in the Raman spectra at ~600 cm⁻¹ corresponds to R6G peak but not the local vibrational mode of Mo-O bond. More detailed information has been provided in the revised manuscript, and also listed as follows:

Supplementary Table 3 | Raman bands for R6G (J. Phys. Chem. 1984, 88, 5935-5944).

Frequency cm ⁻¹	Assignment
612	C-C-C ring in-plane bend
773	C-H out-of-plane bend
1131	C-H in-plane bend
1187	C-C stretching
1312	stretching modes
1360	aromatic C-C stretching
1508	aromatic C-C stretching
1532	aromatic C-C stretching
1575	aromatic C-C stretching
1650	aromatic C-C stretching

Supplementary Figure 16 | Raman spectra of R6G, pristine MoS₂ without R6G

loading and partially-oxidized MoS₂ samples without R6G loading. MoS₂ structure can be confirmed by the appearance of two distinct peaks at 377 and 404 cm⁻¹ for the E_{2g}¹ and A_{1g} vibrational modes, respectively (Nature Materials 2012, 11, 963-969). In contrast, partially-oxidized MoS₂ samples show dramatically altered Raman spectra with several additional bands when compared to pristine MoS₂, indicating the successful incorporation of oxygen atoms. Besides the characteristic bands at 377 and 404 cm⁻¹ for MoS₂, the bands at 112, 123, 146, 195 cm⁻¹ can be ascribed to MoO₂ (Nature Communications 2017, 8, 14903), and the bands at 212, 237, 336 cm⁻¹ are related to MoO₃ (Tribology Letters 2011, 42, 301-310). The broad band at 284 cm⁻¹ (B_{2g}, B_{3g}) is a doublet comprised of wagging modes of the terminal oxygen atoms, while the 664 cm⁻¹ (B_{2g}, B_{3g}) is an asymmetric stretching of the Mo–O–Mo bridge along the c axis, and the intense band at 819 cm⁻¹ (A_g, B_{1g}) is a symmetric stretch of the terminal oxygen atoms, with another intense band at 993 cm⁻¹ (A_g, B_{1g}) being the asymmetric stretch of the terminal oxygen atoms (Tribology Letters 2011, 42, 301-310). It is found that the relative intensities of the Raman bands of these oxygen-incorporated samples could vary significantly as a function of treatment temperature; however, there is no frequency shift on these bands. It is also important to note that the Raman bands of the samples at low-temperature treatment (MoS₂-200 °C and MoS₂-300 °C) indicate the coexistence of Mo-S and Mo-O bonds, while for MoS₂-350 °C sample after high-temperature treatment, the disappearance of A_{1g} (405 cm⁻¹) band evidences a decreased amount of S component as well as the phase transition from pristine MoS₂ towards fully-oxidized MoO₃.

Comment 4:

Abstract - should spell out "LOD" in full

Author reply: Thanks for the referee's comments. We have added a full expression for this.

“..., but also endow MoS₂ with low limit of detection (LOD) below 10⁻⁷ M.”

Comment 5:

p5 lines 110-111

"The significant difference between substitution and oxidation is Mo(VI) can be observed in partial oxidation process but not in oxygen substitution process"

I misunderstood this sentence at first; I think it might help to say "is that Mo(VI) can be observed in the partial oxidation process but not in the oxygen substitution process" - authors please confirm!

Author reply: Thanks for the referee's comments. We have rewritten this sentence as follows: "The significant difference between substitution and oxidation is that Mo(VI) can be observed in the partial oxidation process but not in the oxygen substitution process."

Comment 6:

p6 line 114

"hydrothermal treatment of $(\text{NH}_4)_6\text{Mo}_7\text{O}_{24}\cdot 4\text{H}_2\text{O}$ is performed"

Does this mean "treatment of thiourea in (NH_4) .."? I think this is important enough to be absolutely clear in the main text even if it's in the SI. Same comment applies to the figure 1 caption.

Author reply: Thanks for the referee's comments. We have made it clear in both the context and the caption of Figure 1.

"Low-temperature hydrothermal treatment of $(\text{NH}_4)_6\text{Mo}_7\text{O}_{24}\cdot 4\text{H}_2\text{O}$ with the assistance of thiourea as an additive realize oxygen substitution."

Comment 7:

p14 line 303

"exhilaratingly"

This is perhaps over the top! I think it's the first time in my career that I've seen this word in a scientific paper so, whilst I appreciated the authors' attempt to use interesting language, I'd tone it down a bit.

Author reply: Thanks for the referee's suggestion. We have corrected for this as follows: "Attractively, our approach is generically applicable to other types of

non-metal-oxide semiconductors such as WS₂ and MoSe₂.”

Comment 8:

p19 line 388

"through Fermi's golden rule"

Not sure why it's necessary to mention FGR here - it's such a general principle that it doesn't add any information to say this.

Author reply: Thanks for the referee's suggestion. We have thoroughly reconstructed the discussion section of mechanism studies in the manuscript, and this sentence has been removed.

Reviewers' Comments:

Reviewer #1 (Remarks to the Author):

The authors have adequately addressed my concerns, and I believe this article is now ready for publication.

Reviewer #4 (Remarks to the Author):

I have read the authors' responses to all the reviewers' comments and am happy that the authors have given all comments serious consideration and have responded thoroughly and satisfactorily. The paper is much stronger as a result and I still believe (despite the existence of other, related studies) that this work has sufficient novelty and potential interest for the community to warrant publication in Nat. Comms.